# Mind the Gap: Misalignment Between Drought Monitoring and Community Realities

Sarra Kchouk[1], Louise Cavalcante[2], Lieke A. Melsen[3], David W. Walker[1], Germano Ribeiro Neto[3], Rubens Gondim[4], Wouter J. Smolenaars[5] and Pieter R. van Oel[1]

[1]Water Resources Management Group, Wageningen University, Wageningen, 6708PB, The Netherlands
[2]Public Administration and Policy Group, Wageningen University, Wageningen, 6706KN, The Netherlands
[3]Hydrology and Quantitative Water Management Group, Wageningen University, Wageningen, 6708PB,TheNetherlands
[4]Embrapa Agroindústria Tropical, Fortaleza, 60511-110, Brazil
[5]Water Systems and Global Change Group, Wageningen University, Wageningen, 6708PB, The Netherlands

*Correspondence to*: Sarra Kchouk (sarra.kchouk@wur.nl)

**Abstract**

Despite recent studies emphasising the dual human and physical nature of droughts, there is a lag in advancing this insight in drought monitoring and early warning systems (DEWS). These systems mainly depend on hydro-climatic indices and often overlook the experiences of affected communities, resulting in a drought-monitoring gap. This study introduces the Monitoring Efficacy Matrix (MEM) to assess the alignment between officially monitored data, relevant to drought impacts, and the actual experiences of a rural community in Northeast Brazil, which we investigated through interviews. The MEM revealed 'drought-monitoring challenges', composed of mismatches and blindspots between the official data and local experiences. Mismatches stem from varying spatial and temporal levels; blindspots arise from the diversity of local resilience strategies, or vulnerabilities, influencing drought impacts. What we define as a 'drought-monitoring gap' results from the tendency to prioritise specific indices and pragmatic spatial and temporal levels over a comprehensive drought-monitoring approach. We posit that a first step to bridge this gap can draw inspiration from recent drought-impact-monitoring initiatives, which are focused on the continuous monitoring of non-extreme events by municipal technical extension officers. However, ultimately bridging the drought-monitoring gap remains conditional on the adaptation of DEWS frameworks to accommodate the integration of qualitative and local data representing the relevant drought-related local context.

## 1 Introduction

More and more studies highlight the human influence on droughts, demonstrating that drought results from both natural and anthropogenic drivers (Aghakouchak et al., 2021; Van Loon et al., 2016; Walker et al., 2022; Di Baldassarre et al., 2018) and affects the hydrological cycle and human populations (Savelli et al., 2021; Ribeiro Neto et al., 2022; Kchouk et al., 2023a). Despite this recognition, limited progress has been made in incorporating this knowledge into drought monitoring and early warning systems (DEWSs). DEWSs still predominantly rely on hydro-climatic indices, focusing only on some aspects of drought risk. According to the IPCC framework, drought risk is the interaction of hazard, exposure, and vulnerability (Carrão et al., 2016; UNDRR, 2021). Drought hazard refers to the failure of the system that maintains the hydrological balance, which can include e.g. reduced rainfall over a certain period, inadequate timing or ineffectiveness of precipitation, or a negative water balance due to increased atmospheric water demand from high temperatures or strong winds (UNDRR, 2021). Exposure involves the elements within a system—such as assets, infrastructure, species, ecosystems, and people—that could be adversely affected by the drought hazard (UNDRR, 2024). Vulnerability encompasses the physical, social, economic, and environmental factors that increase the susceptibility of these elements to drought impacts (IPCC, 2014; UNDRR, 2021).

DEWSs often lack indices that account for the human dimensions of drought, specifically: (i) human influences on drought risk and (ii) the impacts of drought on human populations. One reason for the focus on hydro-climatic factors in drought monitoring is the intertwining of physical and anthropogenic aspects within drought risk, making it challenging to quantify and attribute anthropogenic

contributions precisely (Aghakouchak et al., 2021). Anthropogenic drivers add complexity and variability, as they are dynamic and non-static, which complicates their integration into monitoring systems (Kchouk et al., 2023a). Similarly, monitoring drought impacts is difficult because they are non-structural, difficult to quantify or monetise, and can be direct or indirect (Bachmair et al., 2016; Kchouk et al., 2023a; Logar and Van Den Bergh, 2013; Wilhite et al., 2007).

The lack of accounting for human drivers of droughts and their impacts in DEWSs results in what we call a "drought-monitoring gap":

a gap between the monitored data and the drought conditions experienced by human populations. This oversight hinders DEWS from fully achieving their goals of facilitating proactive, well-informed decision-making and providing vulnerable groups with timely, reliable data (Pulwarty and Verdin, 2013; UNDRR, 2021). The local context's specificity adds to this challenge, as understanding and monitoring these human dimensions are complex and context-dependent (UNDRR, 2021). This gap complicates drought managers' ability to make informed decisions and take appropriate action, especially when the anthropogenic drivers of drought risks and impacts remain unknown

outside the affected area. Although there are promising drought-impact-monitoring initiatives (EDII, 2023; EM-DAT, 2023), these efforts are still in the early stages and have mainly enhanced our understanding of drought risk and its impacts (Lackstrom et al., 2022; Stephan et al., 2021; Tijdeman et al., 2022). While this progress is an important step forward, further work is needed to build on this foundation—particularly by incorporating people's lived experiences of drought impacts—to develop actionable strategies that address underlying vulnerabilities and build resilience to future droughts.

Therefore, there is a necessity to address the drought-monitoring gap from both ends: the relevance of monitoring human drivers of drought, and of the drought impacts as experienced by populations. Firstly, monitoring human drivers of drought is important because human actions can significantly influence exposure and vulnerability to drought, impacting both its severity and the effectiveness of mitigation efforts (Aghakouchak et al., 2021; Carrão et al., 2016; Haile et al., 2020; Meza et al., 2020; Van Loon et al., 2016). Walker et al. (2022) detail numerous examples of water and drought mismanagement that led to inadequately addressing or even aggravating

drought impacts. This mismanagement generally resulted from a narrow understanding of the drought threat limited to hydrometeorology. Guidance literature from the Integrated Drought Management Program and others has for many years urged a shift from crisis management to risk management, from costly, ineffective, poorly coordinated, poorly targeted reactive "solutions" to investment in building resilience by addressing the root causes of vulnerability to drought impacts (e.g. IDMP, 2014, 2017; Wilhite, 2000). Secondly, it is important to consider drought impacts and their integration in early-warning or monitoring systems because impact

data improves understanding of vulnerabilities, aids in developing mitigation strategies, supports targeted relief allocation, informs policy, and reflects actual conditions better than hydrometeorological data alone (Walker et al., 2024). This enhanced understanding is crucial for accurate decision-making and resource management in diverse local systems and sectors affected by drought (Wilhite et al., 2007; Hayes et al., 2011; Lackstrom et al., 2013). These reasons have led to drought impact monitoring being referred to as the "missing piece" in drought monitoring and forecasting (Lackstrom et al., 2013; Walker et al., 2024).

In this study, we seek to answer the following question: how can we bridge the existing drought-monitoring gap between the available drought-relevant data that are formally monitored and actual drought impacts, as experienced and reported by local populations? We address this question by focusing on the case study of a rural community in semi-arid, drought-prone Northeast Brazil. Our study aims to compare the drought impacts experienced over time by this rural community with the drought-relevant data formally monitored, covering that same area and also available at different spatial and temporal levels. This comparison was made using our newly developed

Monitoring Efficacy Matrix (MEM), a conceptual tool designed to evaluate the efficacy of drought indices in tracking drought impacts.

The MEM allowed us to identify instances where the two datasets – rural experiences and official data – did not align. We termed these instances 'drought-monitoring challenges'. By examining these drought monitoring challenges and understanding the reasons underlying the drought-monitoring gap, we reflect on whether drought-impact indices –local, contextual, yet replicable and useful for drought (impact) monitoring – are a realistic goal.

## 2    Methods and data

### 2.1    Methodological approach and framework

Our methodological approach comprised three steps.

- Step 1: We explored the drought conditions and impacts experienced over time by the rural population. We focused on the community of Olho d'Água located within the municipality of Piquet Carneiro (Figure 1). For this purpose, we conducted interviews; this approach is detailed further in Section 2.2.

- Step 2: We examined conventional drought indices and officially monitored data relevant to drought impacts that could characterise drought conditions in the focus area. "Conventional indices" refer to commonly used metrics to quantify and characterise drought conditions. These include time series characterising rainfall and meteorological drought (Standardised Precipitation Indices (SPIs), Brazilian Drought Monitor Map), agriculture (cropped and harvested areas, crop yields, agricultural output), and hydrology (reservoir volumes and water surface area). These drought indices and official data are among the most widely used and agreed-upon tools to monitor and characterise drought severity (Bachmair et al., 2016; Kchouk et al., 2022), and they align with the impacts on livelihood, food, and water security that we aimed to explore. These datasets have specific spatial and temporal monitoring levels, which are not necessarily homogeneous across the different datasets, nor aligned with the levels at which impacts are experienced by populations. Specific information about the data series and data collection is provided in 2.3.

- Step 3: We compared the data of Steps 1 and 2 using a newly developed Monitoring Efficacy Matrix (MEM). The MEM is a conceptual framework that aids in identifying monitoring challenges, which include mismatches and blindspots. We designed this framework to examine the alignment of a drought index with reported impacts. The application of the MEM allowed for the evaluation of the alignment between experiences of drought impacts by the population at the community level, and the conventional indices which are also available for different spatial and temporal levels. The specifics of the MEM, along with the definitions of monitoring challenges are elaborated on in Section 2.4.

### 2.2    Case study and data collection

The study focuses on the Olho d'Água community in Piquet Carneiro, situated in the Banabuiú river basin of the state of Ceará (Figure 1). This rural community comprises fifteen households, with members working either within the agricultural sector or in other sectors, such as public service. At the time of the interviews (from November 2021 to June 2022), income-generating and asset-building activities relied on the water from a relatively small reservoir, officially unmonitored, with a maximum water-surface area reaching 14 hectares. The Brazilian state of Ceará, located in the semi-arid region known as Sertão, has faced consistent drought challenges (UNDRR, 2021). The latest multiannual drought (2012 – 2019), noted for its intensity, deeply affected the region's agriculture. Most impacted were smallholder farmers reliant on rainfed agriculture, who experienced significant crop losses and economic setbacks (Brito et al., 2018; Pontes Filho et al., 2020) as well as compromised water availability and quality (Eakin et al., 2014). The region's annual rainfall averaging 750 mm, predominantly occurring from January to April and its annual evapotranspiration exceeding 2000 mm hinder surface

water storage (Martins and Reis Junior, 2021). In response to these challenges, the government invested heavily in water infrastructure during the 1990s and 2000s (Cavalcante et al., 2022). Additionally, private unmonitored small reservoirs became widespread, sometimes limiting the recharge of larger strategic reservoirs, especially during the severe 2012-2019 drought (Ribeiro Neto et al., 2022).

The distinction between what are colloquially referred to as strategic and non-strategic reservoirs is crucial for understanding the local context and the associated monitoring challenges. Strategic reservoirs are large public infrastructure projects that are systematically monitored by state water agencies, primarily serving urban populations. In contrast, non-strategic reservoirs are smaller reservoirs (under 1 million m³ (Rabelo et al., 2022)) that are typically constructed by rural populations to ensure their water access. While strategic reservoirs are always monitored, non-strategic reservoirs are typically unmonitored, though there may be exceptions, as they fall outside the official state-planned reservoir grid. However, these non-strategic reservoirs remain informally strategic at the local level because the majority of rural communities depend on them for their income-generating activities. As these smaller reservoirs are locally built and managed, they elude the control, maintenance, and monitoring of official agencies. In the municipality of Piquet Carneiro, the 'São José II dam' is the only formal strategic reservoir (Figure 1).

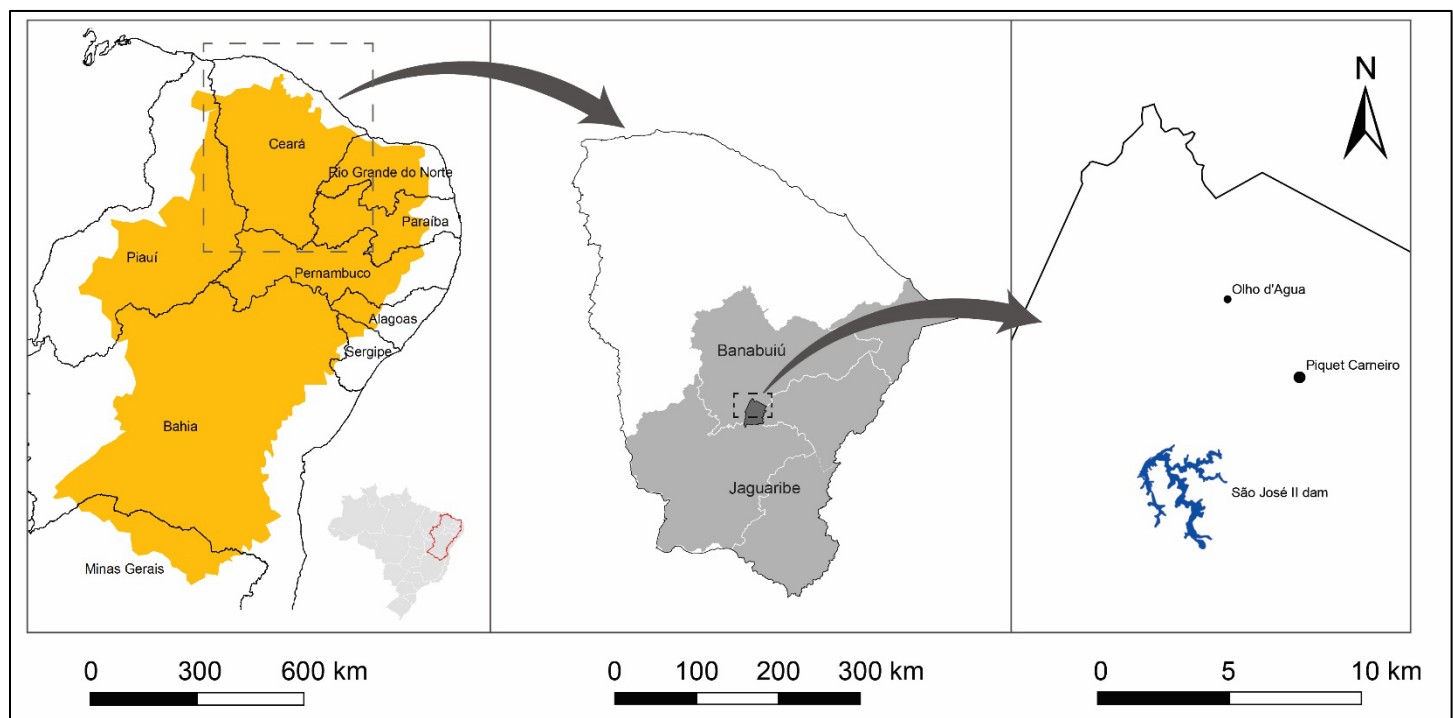

**Figure 1: Map of the case study showing: the semi-arid Northeast of Brazil (left); the state of Ceará, the Banabuiú sub-basin as part of the Jaguaribe River Basin, and the municipality of Piquet Carneiro in dark grey (centre); the city of Piquet Carneiro, the community of Olho d'Água, and the São José II dam in dark blue (right).**

In Piquet Carneiro, fifteen interviews were conducted: eleven with members of the Olho d'Água community and four with practitioners (Table 1). The interviewees and study site were selected through a snowball sampling method, where initial participants recommended other potential interviewees (Parker et al., 2019). The interviews were both unstructured, allowing for open-ended, flexible discussions without predetermined questions, and semi-structured, guided by a set of predefined questions while the rest of the questions were not planned in advance, permitting exploration of other topics in depth (Brinkmann, 2014). After these interviews, no new information emerged, indicating information saturation. These fifteen interviews were part of a more elaborate fieldwork campaign, from November 2021 to June 2022, comprising 41 interviews with farmers and individuals from water and agricultural organizations and covering 12 rural communities in the Jaguaribe River basin (Figure 1). While the fieldwork campaign provided valuable insights that led to further studies focusing on different communities (Ribeiro Neto et al., 2023a; Kchouk et al., 2023a; Walker et al., 2024; Cavalcante et al.,

2023), we focused in this study on the community of Olho d'Água due to its atypical characteristics. Unlike other investigated communities, Olho d'Água has diversified means of water, food, and livelihood security, and did not experience an entire collapse of these systems during or after the multi-annual drought of 2012 to 2019 (see Kchouk et al., 2023a). Therefore, understanding the reasons behind this resilience and whether these factors are being adequately monitored became a key focus.

All interviewees provided consent before being interviewed, in accordance with ethical clearance requirements. The interviews, which lasted between twenty minutes and three hours, were not recorded but were immediately transcribed to ensure the comfort of the interviewees who were predominantly smallholder. None of the solicited individuals refused to be interviewed. Questions were formulated to encourage the participants to describe what they considered to be the drought risks, impacts, and factors increasing or decreasing the likelihood of impactful drought over time in the study area. Table 1 shows how the interviews were conducted.

**Table 1: Interviews Summary**

| Actor | Organisation | Question Themes | Time period referred to |
|---|---|---|---|
| • Practitioners<br>• Rural extension technicians | Agricultural secretaries of Piquet Carneiro | Vulnerable groups<br>Water access<br>Agricultural practices<br>Markets existence and prices<br>Drought impacts<br>Social and agricultural programs and policies | 1970s-2022 |
| • Local government | Municipality of Piquet Carneiro | Vulnerable groups<br>Drought impacts<br>Hydraulic infrastructure<br>Drought emergency state | 1970s-2022 |
| • Farmers and rural inhabitants | Not applicable | Daily life and family<br>Agricultural and livelihood strategies<br>Droughts in the distant and recent past | 1956-2022 |

## 2.3    Drought-relevant data

We extracted data from various international and Brazilian databases (Table 2), covering climatology, reservoir storage, and agricultural production. We used the rainfall data to calculate different Standardised Precipitation Indices (SPIs), each serving a distinct purpose: SPI-3 indicates short-term soil moisture, relevant for crops; SPI-6 provides a mid-term view, affecting agriculture and offering early signs of changes in water storage; SPI-12 monitors long-term trends in water storage and streamflow (WMO, 2012). An SPI of -1 or lower indicates a rainfall deficit, signifying conditions drier than normal across the relevant time scale (EDO, 2020).

Since 2016, the Brazilian Drought Monitor has produced a monthly map of drought conditions, based on drought indices at multiple computation periods from one to twenty-four months. Instead of focusing on a single operational definition of drought—such as agricultural or hydrological—the Drought Monitor integrates indices from different categories to provide a comprehensive understanding of drought conditions. These indices are SPI, Standardised Precipitation and Evapotranspiration Indices (SPEIs), Standardized Runoff and Dry Spell Indicators, and remote-sensing indices, and are validated by regional offices that consider ground observations from networks of observers (De Nys et al., 2016; Walker et al., 2024). The same approach applies to the studied community

of Olho d'Água, located in the municipality of Piquet Carneiro, which is included in the maps produced by the Brazilian Drought Monitor. The Drought Monitor categorises conditions starting from 'no drought' to 'weak drought', which indicate the beginning or end of dry conditions. Categories of 'moderate', 'severe', 'extreme', and culminating in 'exceptional drought', indicate widespread losses in crops and pastures and water shortage at an emergency level. Finally, we obtained agricultural data encompassing the relevant crops in the community of Olho d'Água.

All the utilised datasets, with access links, are available for consultation (Kchouk et al., 2023b).

**Table 2: Step 2 data – conventional drought indices**

| Data source | Information extracted | Time Range |
|---|---|---|
| Climate Hazards Group InfraRed Precipitation with Station data (CHIRPS) | Rainfall time series | 1980-2023 |
| Meteorology and Water Management Institute of the State of Ceará (Funceme) | Small reservoir locations and surface area | 2004-2022 |
| National Company of Water Resources (Cogerh) and Funceme | Sao José II Reservoir volumes | 2004-2022 |
| National Water and Sanitation Agency (ANA) and Funceme | Brazilian Drought Monitor | 2014-2022 |
| Brazilian Institute of Geography and Statistics (IBGE) | Total agricultural production per municipality  Quantity Produced  Crop yield per hectare  Livestock population  Milk and honey production | 1977-2022 |

## 2.4    Monitoring efficacy matrix and drought-monitoring challenges

Monitoring efficacy refers to how effectively a system detects, tracks, and reports on specific parameters or events. In the context of drought, this includes the accuracy, reliability, timeliness, and comprehensiveness of drought indices used by monitoring systems. Drought-monitoring efficacy, therefore, relates to how well these systems detect and report drought conditions, including severity and impacts. Challenges arise when systems fail to capture these conditions accurately, reliably, or comprehensively. We propose a drought-monitoring efficacy matrix to identify and address these challenges.

The Monitoring Efficacy Matrix (MEM) is a conceptual tool designed to evaluate the efficacy of conventional drought indices in tracking various types of drought impacts. It features columns representing conventional drought indices, which are the standardised methods or metrics used to measure and characterise droughts and their conditions. The rows of the MEM classify different drought impacts, organised within and across various distinct levels that subsequently influence the selected impacts (Table 3). By juxtaposing drought indices with these impacts, the MEM provides a comprehensive perspective on how effectively these indices capture the multifaceted impacts of droughts.

**Table 3: Example of an empty monitoring efficacy matrix**

| Scale: e.g. Jurisdictional Level: / Drought indices — Drought impact on E.g.: | | Index 1 e.g.: SPI | Index 2 e.g. Reservoir Volume | … | Index n |
|---|---|---|---|---|---|
| Small E.g. Household | Livelihood | | | | |
| | Food security | | | | |
| | Water security | | | | |
| Middle E.g. Community | Livelihood | | | | |
| | Food security | | | | |
| | Water security | | | | |
| Large E.g. Municipality | Livelihood | | | | |
| | Food security | | | | |
| | Water security | | | | |

The matrix is designed to be a flexible tool for assessing how well drought indices capture the impacts on different systems within a selected spatial level, e.g. rural communities. Each cell in the matrix is intended to be filled with qualitative assessments or specific reasons why an index does or does not accurately reflect the impacts on the focus system. If an index accurately captures the impact, the cell can be filled with a qualitative statement indicating a good match. If not, the cell should detail the reasons for the mismatch or blindspot, which are further elaborated upon in the text as drought monitoring challenges.

The drought indices and official data relevant to drought that we utilised are among the most used and agreed-upon for monitoring and characterising drought severity (Kchouk et al., 2022), and are also related to the impacts we aimed to explore. These indices were selected because they align with the impacts we sought to explore, specifically those related to livelihood, food, and water security at both household and community levels. It is important to note that, in rural communities, these three systems often overlap due to their interconnected nature.

In this study, livelihood security refers to the capacity of households and communities to sustain and enhance their income-generating activities and assets, even in the face of drought. A livelihood comprises the capabilities, assets (including both material and social resources), and activities required to support a living (UNDP and IRP, 2010). Livelihood strategies, which are the combination of activities that people choose or are compelled to undertake to achieve their livelihood goals, naturally evolve as opportunities, risks, and constraints change (Alinovi et al., 2010). For example, income-generating activities such as livestock farming contribute not only to livelihood security but also to food security by providing essential food products like milk, meat, and eggs. Food security focuses on the availability, access, and utilisation of food within households and communities. It considers how drought affects food production, availability, and the ability to purchase or otherwise acquire sufficient food to meet nutritional needs (World Bank, 2024). Similarly, water security is a critical aspect of community resilience, defined holistically by UN Water (2013) as a population's access to adequate quantities of water of acceptable quality. This access is essential for sustaining livelihoods, ensuring human well-being, and supporting socio-economic development (Montanari et al., 2013).

While the overlap between these systems is evident, it does not present a problem; rather, it highlights the interconnectedness of rural livelihoods. The matrix we propose allows for the assessment of these systems individually while acknowledging their interdependencies, thereby providing a comprehensive understanding of the impacts of drought at the community level. Furthermore, the MEM is adaptable and can be modified to include other systems or indicators, such as those affecting e.g. ecosystems, hydroelectric production, health, or market trade, depending on the specific context and needs of the analysis. Scale choices depend on what is to be assessed with the MEM; it must align with the chosen impacts.

Scale refers to the dimensions used to measure and study phenomena, whether they are spatial, temporal, or analytical. Within these scales, levels represent specific units of analysis (Gibson et al., 2000). Spatial levels can for example range from the plot to the basin and time levels can range from seconds to decades; it all depends on the studied phenomena. For example, on a spatial scale, events can range from cellular processes to global climate changes, while on a temporal scale, they can cover rapid events like hurricanes and long-term societal shifts (Cash et al., 2006). Drought and its impacts cover several levels, both at spatial and temporal scales (Kchouk et al., 2023a). Furthermore, it is not only the physical aspect of drought that determines the severity of droughts. Anthropogenic factors, even if indirectly related to drought, can amplify the impacts. For instance, the likelihood of drought affecting the livelihood, water, or food systems also depends on how diversified the considered system is. The more the considered system is reliant on one source, the more likely it is to be impacted by drought and collapse; the more diversified it is, the more resilient to drought impacts, and the less likely it is to face severe impacts (Kchouk et al., 2023a). Thus, adequate drought monitoring should be comprehensive of all the levels within the spatial and temporal scales where the system might be impacted, and also of all the elements within the system that determine its resilience to drought impacts.

Monitoring challenges arise when the drought indices do not comprehensively and accurately capture the impact at the selected level of analysis. Such monitoring challenges fall into two types: mismatches and blindspots. A mismatch occurs when the level at which monitoring takes place (be it the level defined by official data or of a drought index) does not align with the spatial or temporal reach of the impact aimed to be monitored. Blindspots result from not monitoring all the elements that contribute to the resilience or vulnerability of the considered system to drought impacts.

When filling in our MEMs, mismatches and blindspots emerged when real-world experienced impacts were compared with the official data. In our case, these monitoring challenges appeared when we could not find impacts mentioned by the population of Olho d'Água in the official monitoring data. In our study, mismatches and blindspots occur in the following instances:

(i) Mismatches occur when impacts, or signals of these impacts, mentioned by the rural populations cannot be found in the official data because the official data level is too broad or too narrow, either in space or time, to capture the extent of the experienced impact. For example, a spatial-scale mismatch might arise if official livestock data are available at municipality level, counting tens of thousands of cows, while in reality, each individual household within a specific community only owns about five cows. Such data, because of its broad scale, might not accurately depict the experiences of every community within the municipality. A temporal-scale mismatch might emerge for example if a drought indicator's timeframe is too extended to capture shorter, yet impactful, events within its range. An example is the SPI-1, the shortest SPI, which sometimes overlooks impactful flash droughts; because it is based on monthly data, it cannot detect dry spells shorter than a month and thus misses the rapidly developing and intensifying flash droughts that can occur within weeks or even days of precipitation deficits (Walker et al., 2023).

(ii) Blindspots occur when the official data only capture the range of elements composing the considered system in an incomplete or limited manner. This could either lead to an underestimation or an overestimation of vulnerability. For example, a blindspot can occur when small reservoirs, pivotal in many communities' water systems, are only counted rather than having their volumes monitored. Overlooking volumes might lead to overestimating the physical water availability and therefore, underestimating the vulnerability to drought impacts. Another example can be when the livelihood system of a community relies on the sale of very specific cash crops while agricultural monitoring focuses on subsistence crops. Such crucial elements can be overlooked by official data because the monitoring level is too broad to accurately capture them, as these elements are too specific to a limited area or a limited period of time; in other words, blindspots can sometimes be caused by mismatches.

Confronting conventional drought indices with the impacts experienced by rural populations provides insights into what is needed for local and context-specific drought impact indices. Identifying mismatches and blindspots allows us to identify the missing information essential for a comprehensive understanding of drought impacts tailored to particular systems, levels, and local contexts. While our exploration is specific to our case study area (Section 2.2), this study serves as a foundation for assessing the effectiveness of broader-scale monitoring. This study inherently poses the questions of up to what level can we effectively monitor drought and its impacts and if drought impacts indices that are generic and replicable, yet specific to the area, are possible to develop.

## 3    Results

### 3.1    Drought impacts experienced by rural populations of Olho d'Água, Piquet Carneiro

This section offers a summary of the trajectory of the Olho d'Água community to aid understanding of Section 3.3 in which we develop the MEMs. Detailed narratives are in the supplementary material.

**Table 4: Overview of the main elements composing the livelihood, food, and water systems in the community of Olho d'Água over time**

| Period / System | Pre-2003 | 2003-2012 | 2012-2019 |
|---|---|---|---|
| **Livelihood System** | • Rainfed subsistence crops: pastures for livestock, small areas of cotton (max. 1 ha per household). Surplus of beans and maize was sold. | • Honey production<br>• Irrigated and diverse onsite food production (sold at the local market, door-to-door, through the governmental program)<br>• Food processing (sold at the local market, door-to-door, through the governmental program) | • Food processing (from food bought elsewhere) – sold at the local Piquet Carneiro Market and through governmental programs<br>• Honey production<br>• Cash transfers programs |
| **Food System** | • Rainfed subsistence crops (beans and maize)<br>• Milk from livestock (2 cows max. per household) | • Buying from supermarkets<br>• Food produced onsite<br>• Milk from livestock (max. 5 cows per household) | • Buying from supermarkets<br>• Food produced onsite<br>• Milk from livestock (max. 5 cows per household) |
| **Water System** | • One community shallow well (for drinking) | The community shallow well was replaced by:<br>• Individual shallow and deep wells (for irrigation)<br>• Cisterns (2 per household, for drinking and irrigation)<br>• Community's small unmonitored reservoir (for irrigation) | • Cisterns (only for drinking – no more irrigation)<br>• Water trucks (only for drinking)<br>• Wells and small reservoirs dried up |

The earliest recollection of droughts we gathered in Olho d'Água community start in 1958. Until 2003, the livelihood, water and food systems were highly dependent on rainfall (Table 4). Household food consisted of subsistence rainfed maize and beans and milk from two cows maximum per household. The rare surplus would be sold for cash. Some households in the community also had small patches of cotton for selling. The drinking-water system was reliant on a shallow well for the whole community. Until 2003, droughts severely impacted the water, food, and livelihood securities, also aggravated by a lack of alternatives and governmental interventions. Notably, the droughts of 1958 and 1970 led to food and income insecurities, made worse by rising staple prices and depleted community finances. The government's "Workfronts" initiative (Costa, 1974; Rocha, 2001) during this period offered employment but inconsistent payments.

Later droughts, spanning 1983 to 2003, affected household and community water security, with the only community well drying up. The community also suffered food insecurity from crop failure and livestock deaths.

However, from 2003, there was a significant shift in the community's experience of drought impacts due to improved water management and governmental policies. Agriculture diversified from traditional livestock and subsistence crops to beekeeping, fruit production and their onsite processing (Table 4). These three activities have become the main source of agricultural income in the community. Several government programs, like a local beekeeping educational project introduced in 2007 through the Sustainable Development Program for Rural Territories (Programa de Desenvolvimento Sustentável de Territórios Rurais- PRONAT), the Food Acquisition Program (Programa de Aquisição de Alimentos- PAA) and the National School Feeding Program (Programa Nacional de Alimentação Escolar - PNAE), both introduced around 2003, greatly assisted this diversification, enabling greater resilience against drought impacts. These programs supported local agricultural initiatives, encouraging crop and income diversification, and facilitated income stability during the 2012-2019 drought. In addition, more community members sought employment outside of the agricultural sector. The diversification of the agricultural system was also made possible through the community's small reservoir (constructed between 2003 and 2012, though the exact year is not recalled by anyone) and the introduction of cisterns, with each household benefiting from two. Cisterns allow the harvesting of rainwater but can also be filled by water trucks, subsidised by the national government, during periods officially declared as 'emergency situations'. In 2005, households in the community received their first cistern, installed as part of a national government program to provide drinking water security. In 2007, a second, larger cistern was provided to each household, enabling them to use water also for irrigation. Farmers also dug shallow wells in their plots for irrigation.

During the 2012-2019 drought, the community's diversified income sources and proactive interventions, in addition to governmental measures, buffered impacts on the livelihood, food and water systems, avoiding their collapse like in the pre-2003 period. Cattle were still affected, crop yields declined, and all water sources dried up. Livelihoods were maintained from food processing, with food not necessarily produced onsite but bought elsewhere, and with the sale assured through the PAA and PNAE. Honey production, albeit affected, was maintained. Livelihood was also maintained by income from other jobs, receiving crop insurance (Garantia Safra (Kühne, 2020)), and benefiting from a cash transfer programme (Bolsa Família(Soares et al., 2010)). As their livelihood was stable, people could afford to buy food. The local water sources dried up but water trucks were deployed, even though quantities were below what was needed (Table 4). By 2020, the community experienced a recovery in agricultural production due to the replenishment of their reservoir during the rainy season. This recovery triggered farmers to invest in innovative farming techniques, such as hydroponic systems and greenhouses.

A notable challenge to the livelihood system, not related to drought, is the aging population of Piquet Carneiro and their purchasing power. Specifically, retirees, who predominantly purchase farmers' products in local markets, determine the sales pattern. According to census data, the ageing index (*índice de envelhecimento*, used by the IBGE) in Piquet Carneiro increased from 66 to 110.16 between 2010 and 2022, meaning that for every 100 children aged 0 to 14, there were 110.16 adults aged 60 or older by 2022. This shift towards an ageing population is more pronounced in Piquet Carneiro compared to the state of Ceará, where the ageing index increased from 41.56 to 71.6 over the same period (IBGE, 2022). Sales tend to fluctuate, largely because the majority of buyers are retirees, whose purchasing power depends on the timing of their pension payments. Sales generally dip towards the end of the month, coinciding with the period just before pensions are paid. The availability of cash in banks also significantly influences the purchase of farm products. Piquet Carneiro's banks frequently experience cash shortages as retirees withdraw their full pensions concurrently. Some buyers resort to traveling to other cities to withdraw their pensions, capitalising on this trip to buy products from the local markets. The farmers interviewed noted that while some other farmers prefer to sell in these other cities, they choose to remain in Piquet Carneiro. Failing to

make sales in Piquet Carneiro, they don't incur any financial setbacks. However, traveling to another location brings the risk of being at a loss, by incurring fuel expenses without any return if no sales are made.

### 3.2     Conventional and official drought data

We gathered data related to the physical drivers of drought, as well as the most common direct impacts on reservoir storage, and agricultural production, from various sources for Northeast Brazil (Table 2).

Our figures representing agricultural data focus on the most produced crops (Figure 2a), those covering the largest harvested areas, and those yielding the most (Figure 2b). For most crops shown, the areas harvested and cropped are identical. Therefore, we have combined both types of areas into a single axis in Figure 2b and c. Comprehensive graphs, encompassing all crops cultivated within the municipality, can be found in the supplementary material. We also highlight agricultural products that are significant in the farmers' experiences (mentioned in Table 4), such as bananas (Figure 2c), livestock, honey and milk production (Figure 2d). Bananas serve as

cash crops, and their sale contributes to income. On the other hand, staple crops like beans and maize are primarily produced for family consumption, with any surplus being sold. Some crops which appear to be pivotal for contextualising the farmers' narratives are not available in the official agricultural data, perhaps due to their limited scale (e.g. cassava, soursop and guava). The absence of such local key crops is addressed in the subsequent section to highlight a blindspot.

The lower part of Figure 2 below depicts timeseries of SPIs 3, 6, and 12, highlighting periods with below-average rainfall that might

result in droughts. Additionally, starting from 2014—the year the Brazilian Drought Monitor started monthly reporting of drought severity—these figures also show the portion of the municipality impacted by each drought severity level. Figure 3 displays the change in the number of reservoirs larger than 0.5 hectare for the period 2008-2020. Only their counts and locations, through detecting their water surface, are officially monitored and not their volumes.

The quality of the datasets varied depending on their sources (see Table 2 in the Methods Section). Our primary intention was to visually

represent and juxtapose the data with the experiences of the community of Olho d'Água. Our aim is not to evaluate the data quality or identify correlations among meteorological, agricultural, and hydrological data. However, certain discrepancies and contradictions are evident. For instance, the cotton area declines without any apparent replacement (Figure 2b), which is also later addressed to highlight monitoring challenges.

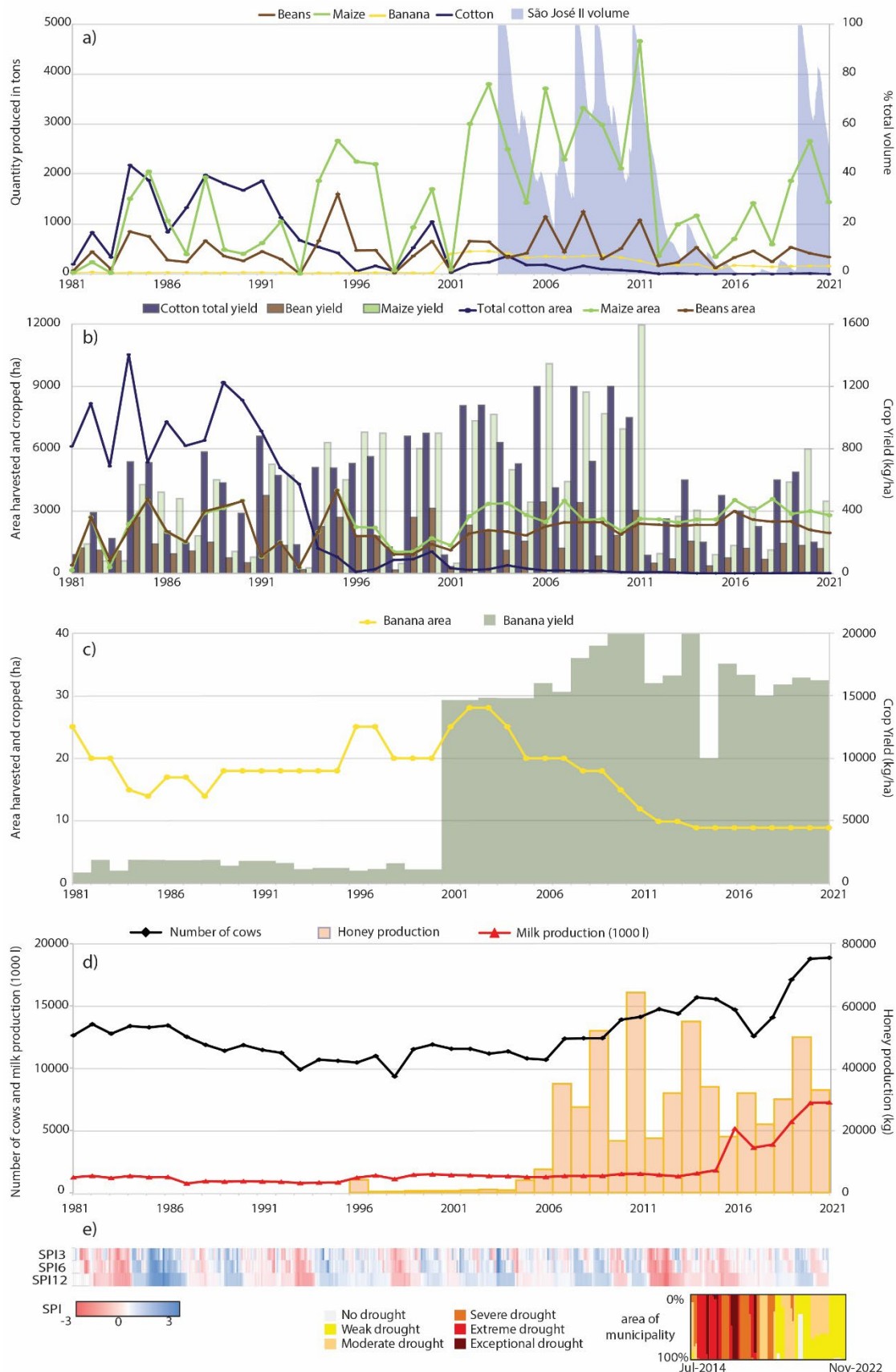

**Figure 2: (a) Annual crop quantity produced in tons in the municipality of Piquet Carneiro from 1974 to 2019 (lines). Daily evolution of the percentage of total volume of the Sao José II dam, which is the only monitored dam in the municipality of Piquet Carneiro (blue shaded area); (b) Annual equal cropped and harvested area in Piquet Carneiro from 1981 to 2021 (lines). The columns represent the annual crop yield per hectare. (Source: IBGE, PAM); (c) Annual equal cropped and harvested area of banana in Piquet Carneiro from 1981 to 2021 (lines). The**

**columns represent the annual crop yield per hectare of banana. (Source: IBGE - PAM); (d) Annual livestock population and production in Piquet Carneiro with the number of cows' head (black line), the annual milk production (in thousands of litres, red line), and the annual honey production (in kg, orange columns. Sources: IBGE - PPM); (e) colour bars of the monthly values of the SPIs 3, 6, and 12. Below are the monthly percentages of the municipality under different categories of drought severity, from July 2014 to November 2022 (white: no drought, yellow: weak drought, light orange: moderate drought, dark orange: severe drought, red: extreme drought, brown: exceptional drought Sources: Cogerh/Funceme, Brazilian Drought Monitor, CHIRPS).**

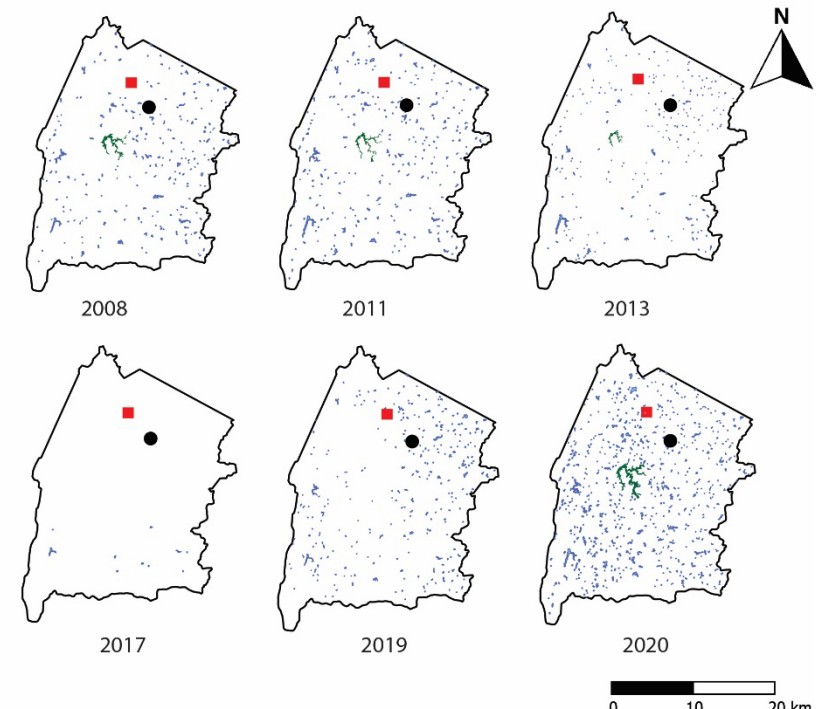

**Figure 3: Maps of the non-strategic reservoirs with a surface area >0.5 hectare in the municipality of Piquet Carneiro from 2008 to 2020 and between the months of July and November. The only officially monitored and strategic dam, Sao José II, is in dark green. The capital Piquet Carneiro is the red square and the community Olho d'Água is the black circle (Funceme, 2020).**

The Standardized Precipitation Indices (SPIs) highlight various meteorological drought events (Figure 2e). Between 2003 and 2012, no severe meteorological drought events were indicated, especially when contrasted with the preceding and succeeding decades. From 1981 to 2003, four multiannual meteorological drought events occurred. From 2012 onwards, a multi-annual drought persisted until 2019. During this period, monthly drought maps produced by the Brazilian Drought Monitor began to be elaborated, categorising the percentage of the municipality affected by different severities of drought.

Before 2003, surface water data were quite limited, with neither strategic nor non-strategic reservoirs being closely monitored. Monitoring data for the São José II dam began in 2004, and for small "non-strategic" reservoirs in 2008. From 2004 to 2012, Sao José II dam monitoring showed significant volume dips that did not consistently align with drought periods. Notably, the Sao José II reservoir dried up entirely in 2017 but regained its maximum capacity by 2020 (Figure 2a). As for non-strategic reservoirs, data are available every one to two years and inform us about their location and water area, as long as it exceeded 0.5 hectare. From 2008 to 2011, the count of small reservoirs increased. Then, from 2012 to 2019, their number began to decline, hitting a low point in 2017 before rebounding in 2019 to numbers higher than before the drought (Figure 3).

The agricultural data up to 2003 highlight cotton as the dominant crop in both quantity and area within the municipality (Figure 2b and c). This dominance saw a sharp decline around 1995, coinciding with periods of low rainfall. Interestingly, the agricultural data show that cotton yield continued to increase until 2012, even though its cropped area and produced quantity were nil. Starting in the early 2000s, maize production saw a significant increase, but its occupied area remained constant. Between 2003 and 2012, yields of both

maize and bananas increased, despite no corresponding growth in their cropped areas. From 2012 to 2019, the area allocated to staples like beans and maize stayed the same, yet their yields and quantities declined.

Regarding livestock data (Figure 2d), there was a consistent decline in cattle numbers prior to 2003. From 2003 to 2012, the number of livestock steadily increased, although milk production remained stable. Starting in 2008, honey production began to rise in the municipality, experiencing fluctuations with some years showing up to three times more honey production than others. After 2012, cattle numbers continued to increase until experiencing a decline in 2017. Milk production, however, remained relatively stable during this time. Starting in 2018, cattle numbers began to rise again, reaching their highest levels ever by 2022. Interestingly, milk production saw

a five-fold increase from 2015 to 2016, dipped slightly in 2016 and 2017, and then surged to its highest levels from 2018 onward, following the pattern of livestock numbers.

### 3.3     Confronting experienced impacts and conventional drought indices in the Monitoring Efficacy Matrix

We completed three MEMs for three different periods: pre-2003 (Table 5A), 2003 to 2012 (Table 5B), and 2012 onwards (Table 5A); the three periods were selected due to their differing contexts. What changed were management practices introduced in the meantime

that later alleviated or worsened drought impacts. By examining the MEM, we aim to understand the reasons underlying the potential monitoring challenges. By comparing the three MEMs, and the monitoring challenges, we aim to understand what information is still lacking for well-informed drought management. We identified a multiplicity of mismatches and blindspots, varying per time period. Despite these variations, there are similarities in these monitoring challenges, and patterns do emerge. All these challenges are compiled in an overview table (Table 5D), which summarises the main types of monitoring challenges. These challenges are further elaborated

on following Table 5.

**Table 5: Monitoring efficiency matrices over the three different periods: Pre-2003 (A), 2003 to 2012 (B), and 2012 to 2019 (C); and monitoring challenges overview (D). Monitoring challenges include 'mismatches' (indicated by the letter M) and 'blindspots' (indicated by the letter B).**

**Table 5A: Pre-2003**

| Impact on | SPI (0.05°, monthly, values below -1) | % of municipality in each drought severity category | Water surface area > 0.5 ha | Reservoir level | Quantity produced (municipality, annual) | Area harvested (municipality, annual) | Crop yield/ha (municipality, annual) | Livestock (municipality, annual) | Milk (municipality, annual) | Honey (municipality, annual) |
|---|---|---|---|---|---|---|---|---|---|---|
| Livelihood security (LS) | SPI (0.05°, monthly, values below -1) | Blindspot as not monitored | N/A as community did not use reservoir storage | | M0 & M7 ; B8 | M0 & M7 ; B8 | M0 | M1 | M1 | N/A as community did not produce honey |
| Food security (FS) | Match. FS extremely dependent on rainfed subsistence crop | | | | B8 | Match, with the overfocus on beans and maize. Crashes of the staples in years of drought | M7 | M1 | M1 | |
| Water security (WS) | Match. Shallow wells extremely dependent on rainfall | | | | M7 | M7 | M7 | M1 | M1 | |

**Table 5B: 2003-2012**

| Impact on | SPI (0.05°, monthly, values below -1) | % of municipality in each drought severity category | Water surface area > 0.5 ha | Reservoir level | Quantity produced (municipality, annual) | Area harvested (municipality, annual) | Crop yield/ha (municipality, annual) | Livestock (municipality, annual) | Milk (municipality, annual) | Honey (municipality, annual) |
|---|---|---|---|---|---|---|---|---|---|---|
| Livelihood security (LS) | B9 | Blindspot as not monitored | B10 | M3 | M5 / B9 | B9 | M4 | M1 / B9 | Match. Milk production is stable | M2 |
| Food security (FS) | B9 | | B10 | M3 | M5 / B9 | B8 | B11 | B9 | B9 | B9 |
| Water security (WS) | Match. No long droughts affecting WS | | B10 | M3 | B11 | Match. The Stable trend suggests water security | B11 | Match, as the increasing trend suggests water security | Match, as the stable trend suggests water security | N/A |

**Table 5C: 2012-2019**

| Impact on | SPI (0.05°, monthly, values below -1) | % of municipality in each drought severity category | Water surface area > 0.5 ha | Reservoir level | Quantity produced (municipality, annual) | Area harvested (municipality, annual) | Crop yield/ha (municipality, annual) | Livestock (municipality, annual) | Milk (municipality, annual) | Honey (municipality, annual) |
|---|---|---|---|---|---|---|---|---|---|---|
| Livelihood security (LS) | B9 & B12 | M6 | B10 | M3 | B8 & B9 / M9 | B8 & B9 / M4 | M4 | M1 | M1 | M2 |
| Food security (FS) | B9 | M6 | B10 | M3 | B9 / M9 | B9 | Match | B9 | B9 | B9 |
| Water security (WS) | B9 | M6 | Match. Reservoir dry | M3 | Match: decrease in trends suggests water insecurity | M7 | B11 | M1 | M1 | N/A |

**Table 5D: Monitoring Challenges Overview**

**Mismatches: the spatial or temporal level of the official monitored data masks locally experienced drought impacts**

M0: Mismatch in terms of whether larger-scale cotton data can be accurately applied to the community.

M1: Mismatch in terms of whether larger-scale livestock data can be accurately applied to small-scale cattle farming in the community.

M2: Mismatch in terms of whether larger-scale honey production data can be accurately applied to the honey production in the community.

M3: The reservoir is not used by the community of Olho d'Água. It is not the appropriate indicator for livelihood, food, or water security at the community level.

M4: High yields paired with limited cropped area in the data at the municipality level actually suggest a focus on a single farm, challenging its generalisability across the entire municipality.

M5: A temporal mismatch emerges when the chosen time scale for monitoring does not capture the actual duration over which events or impacts unfold, making it difficult to accurately assess their influence on LS, FS, or WS within that specific time frame.

M6: Since the monthly drought map is produced at almost national scale and refined to sub-state scale, there is a mismatch: the Drought Monitor was not designed to show the drought severity classification at even municipality scale, never mind for individual communities. When there is variation within the state, we cannot be sure which communities fall under which drought category and how this concretely affects their LS, FS, and WS.

M7: Data stable or upward trend suggests irrigation, which was not the case as the community declared water insecurity and the impossibility of irrigating. This suggests that the monitoring might have been focusing on a single farm irrigating, thereby challenging its generalisability across the entire municipality.

**Blindspots: not all the elements of resilience or risk of the systems to drought impacts are monitored**

B8: Blindspot on the full scope of agricultural practices that support livelihood and food security in the community.

B9: Blindspot on the full scope of alternatives to rainfed agriculture, or alternatives to the variable monitored by the index, that prevent the collapse of the LS, FS, and WS.

B10: Blindspot emerging from important components of the indicator, the water volume not being accounted for.

B11: Blindspots related to the practices of irrigation. The trend of the monitored variable suggests irrigation, for which there is a blindspot as we do not have any official monitoring data in that regard.

B12: A blindspot emerges because the crucial information regarding cash shortages, which affect market sales and consequently livelihoods, is not considered at all.

**Before 2003: Low community resilience to drought, with unreliable and incomplete monitoring data.**

The community's resilience to drought impacts is low due to heavy reliance on rain for livelihood, food, and water. This results in severe impacts during droughts, and the SPI matches these drought periods as described by rural populations. During this time, there are no available hydrological data for areas smaller than 0.5 ha, creating a blindspot given the community's reliance on shallow wells for water security before 2003. Agricultural monitoring omits pastures crucial for the community (Blindspot 8). Contrastingly, one interviewee who lived in the community during the peak of cotton production in Ceará in the 1960s and 1970s, mentioned that cotton production was not prominent in the community. Thus, the agricultural data (Figure 2a and b) may inaccurately emphasise cotton's prominence in the area (Mismatch 1) and its high yield contrasts with its scant production in the community before stopping. The stable trend in cotton during droughts also suggests irrigation, in contrast to the community's water scarcity experiences reporting full loss and stating that they were not irrigating prior to 2003 (Blindspot 11). Municipality-level livestock data, ranging from 9000 to 13500 cows, is not comparable to the community's owning not more than two cows per household, nor applicable to the milk production within the community (Mismatch 1).

**Between 2003 and 2012: Increased diversification in livelihood and food systems, with partial but still inadequate drought monitoring.**

During this period, no multi-annual droughts occurred. As previously stated, livelihood, food, and water systems have diversified. Consequently, while rainfall previously exerted a strong influence on each of these systems, rainfall alone cannot explain current impacts anymore because the resilience of the systems to drought has increased. This is also true for the other indicators. The community's livelihood is not exclusively dependent on onsite food production or agriculture anymore, given that more individuals now work outside this sector. Thus, the SPIs only offer a partial view of the resilience of the livelihood system (Blindspot 9). Moreover, stable incomes ensure food security, which is no longer solely linked to subsistence farming as in the past (Blindspot 8). The reservoir level is not representative for the community that does not utilise it (Mismatch 3). Small reservoirs are crucial, hinting at usage patterns in communities. Yet, monitoring of these reservoirs is incomplete as their volumes or levels are not officially monitored or available (Blindspot 10). The stable trend of cropped areas, coupled with increased production, especially of banana and maize, suggests irrigation practices. However, we lack data on irrigation, which is a crucial element of water security (Blindspot 11). The rising livestock trend in the official data, ranging from 12000 to 15000 cows, does not reflect community patterns, with households owning no more than five animals (Mismatch 1). Similarly, honey production remains predominantly a household activity, even though it is the primary source of agricultural income in the community. While the data show fluctuation of honey production, the community reported only increases. Therefore, it is also challenging to apply such data to the community level (Mismatch 2). Also, looking at the agricultural production of one year is not conclusive to evaluate whether the community was livelihood or food (in)secure during that year. Families generally store part of a year's production, for consumption, processing, or sale in other years when the production falls short (Mismatch 5).

**From 2012 onwards: Greater resilience to drought due to alternative measures, yet continued monitoring challenges**

However, during the prior decade, the community developed or benefitted from resilience mechanisms for their water, food, and livelihood systems. These mechanisms also remained robust as they were not weakened by any severe droughts during that period. Therefore, despite the 2012-2019 drought affected the livelihood, food, and water systems, they were not as severely impacted as they were before 2003, because of alternative governmental measures like *Bolsa Família*, *Garantia Safra*, PAA, PNAE, and water trucks. These alternatives are not accounted for or officially monitored (Blindspot 9). During the drought, the Brazilian Drought Monitor produced monthly maps from which the percentage of the municipality under different categories of drought severity can be extracted. However, it remained unclear under which categories the rural communities fell or what these categories implied in terms of impacts on

water, food and livelihood securities (Mismatch 6). The stable cropping area suggests ongoing irrigation, but this is not the case as the community reduced its cropped area, or even eliminated the banana production, and had to stop irrigation (Mismatch 7).

Furthermore, the reported high yield of bananas, considering the limited cultivated area, raises questions about its accuracy and its generalisability to other communities (Mismatch 4). The quantity of basic staples such as beans and maize decreased during the drought (Figure 2a), leading to the surplus from previous years to be fully consumed in the initial years of the drought. The food security of the community did not depend on these staples anymore as they were income-secure and could afford to buy food produced elsewhere, but this shows how some impacts can manifest long after the time they are monitored (Mismatch 5). The fluctuating honey production,

shown in Figure 2d, might not accurately reflect the community's situation. Honey production, the main source of agricultural income, declined significantly in the community during this period and recovered only in 2020. This suggests a mismatch in the applicability of larger-scale data to the honey production trend in the community (Mismatch 2). The same mismatch is evident in the livestock data, where a trend showing 12,000 to 20,000 cows is too broad to reflect the local average of five cows per household (Mismatch 1). Additionally, factors like cash shortages in local banks, which are not related to drought but affect farmers' income, are not being

monitored (Blindspot 12).

## 4 Discussion

### 4.1 Implications for drought monitoring at community level in Northeast Brazil

The focus of our research on a small rural community in Northeast Brazil is useful to underscore a crucial point for drought monitoring: it is imperative to understand how the focus system is impacted by drought in order to monitor drought impacts efficaciously. We have

previously advocated for a system-oriented and contextualised perspective in drought monitoring (Kchouk et al., 2022; Kchouk et al., 2023), where the considered systems represent components of human welfare that are affected by drought. In this study, we have taken livelihood, food and water securities as focal systems, and examined how they have been impacted at the community level differently over time by different drought events, as the local context changed. We have assessed if drought impacts were effectively captured by conventional drought indices and official data. Such comparison was made using a newly developed Monitoring Efficacy Matrix (MEM)

and aimed to detect drought monitoring challenges, consisting of mismatches and blindspots. Mismatches draw attention to the misalignment between spatial and temporal levels of monitoring and the experienced drought-impacts, while blindspots point to the absence of monitoring of all elements of drought risk and resilience of the focus system. As systems undergo transitions, like the transition from substantial to more diversified agriculture, these elements also change. Therefore, what needs to be monitored evolves as well, reinforcing the necessity for a systems perspective in drought monitoring, rather than the current hydroclimatic-oriented

approach.

Our findings support this always-evolving system perspective. The three MEMs revealed monitoring challenges that were different for the three different time periods. Over these three distinct and consecutive periods, the efficacy of drought monitoring appears to decrease as the community's livelihood, food and water systems diversified and became more resilient. During the first period, when the community was still largely dependent on rainfall, the monitoring aligned reasonably well with experienced drought impacts, although

it remained incomplete. In the following periods, as the community diversified its livelihood, food, and water sources, the monitoring gap also increased. This indicates that as systems became more complex and resilient, conventional indices and data became less capable of capturing the entire range of nuances of that resilience to drought impacts. Some blindspots can be caused by monitoring systems not accounting for all or some aspects of the resilience to drought impacts. Some examples include overlooking alternative income sources, community reservoirs' volumes, the influence of government programmes, or cash shortages caused by a population mainly comprising

retirees. Such blindspots occur due to the plurality of perspectives on what constitutes the livelihood, food, and water systems and what constitutes their resilience to drought impacts, or in simple terms "what should be monitored and how?" This plurality of perspectives is discussed further in the next section.

Additionally, mismatches can also arise from the misalignment between the scales and levels at which conventional drought indices are available and the scales and levels at which impacts are actually experienced. Such mismatches can be temporal, occurring when the

chosen timeframe for monitoring does not align with the duration or frequency of impacts or mitigation strategies. They can be spatial when aggregated, large-scale data do not accurately reflect smaller-scale, local conditions. Spatial mismatches can also occur in the other way around, when data is too specific and mostly skewed by outliers, reducing its applicability at a larger level. Such mismatches occur due to the plurality of scales and levels at which drought drivers and impacts can or should be monitored. This plurality of monitoring scales and levels is also further discussed in the next section.

**4.2     Reflections on what this analysis reveals about drought monitoring**

The term "plurality" is commonly used in the literature on scales and levels (Cash et al., 2006; Wiegant et al., 2020; Poteete, 2012). Plurality refers to the failure to recognise heterogeneity in the way that scales are perceived and valued by different actors, even at the same level. This challenge surfaces when there is an assumption of a single, universally suitable characterisation of scale and level for the entire system or all actors. In this present study, this plurality of scales is characterised by the different mismatches, highlighting the

impossibility of detecting locally experienced impacts, mentioned by the population, as the monitoring data does not cover the spatial or temporal reach of these impacts.

We believe that the concept of 'plurality' can be broadened to cover the heterogeneity of perspectives on livelihood, water, food security, or any other component of human welfare and what characterises this component. The challenge can emerge from assuming that a specific system holds higher importance or priority unanimously for all involved actors. For instance, one might assume that for everyone

involved in drought management, water security is the primary concern. Another assumption might be that the elements that make up a system are consistent for all spatial, temporal and jurisdictional scales. For example, assuming that all rural communities in a municipality rely mainly on rainfed subsistence agriculture. Drought monitoring faces this challenge of plurality as it often standardises both scales and perspectives of impacts. Yet, this study and others in the literature highlight the varied spatial and temporal reach of drought impacts, as well as the varied nature of these impacts, the range of people they affect, and how these impacts also vary according

to the actors impacted (Van Oel et al., 2019; Savelli et al., 2021; Kchouk et al., 2022).

The reasons behind the oversimplification of scales and perspectives in drought monitoring can be traced back to its purpose: to inform and guide decision-making. Three interconnected reasons can explain this standardisation: (i) stakeholders' varied interests; (ii) control; and (iii) simplification (Cash et al., 2006). (i) The way issues are defined in terms of scale often aligns with varied stakeholders' goals and interests. This is because defining the scale of a problem determines who makes decisions and who benefits from them, with the

risk of sometimes resulting in unequal outcomes. For instance, (Van Oel et al., 2019) pointed out that water-for-food governance encompasses multi-level actors, each with different perspectives and impacted differently by drought, therefore necessitating different indices of drought impacts. This leads to (ii) control, through governments framing problems (Van Lieshout et al., 2011), including droughts, to fit within their jurisdiction in their bid to manage issues within their reach and mandate. For example, a government or authority might use a specific indicator to assess drought severity across a jurisdiction, even when the severity can differ considerably

within that area. This approach allows governments to standardise their responses and resource allocation according to predefined administrative boundaries. A perfect example to illustrate this case is the *Garantia Sáfra* – the Index-Based insurance mentioned earlier in this study (Section 3.1). In case of droughts or heavy rains, agriculture extension officers visit selected fields and assess whether crop

losses exceed fifty percent. Pay-outs to the whole region occur if the 'Water Requirement Satisfaction Index', in the respective municipality is reached (Kühne, 2020). Drought monitoring can be reduced to a particular scale, level, or perspective for (iii) simplifying drought management. This is why drought management tends to be siloed across different ministries, departments, or authorities (Wilhite, 2019), due to its different effects on virtually all aspects of society (Bressers et al., 2016). This siloing can in turn complicate drought governance by fragmenting the responsibilities of drought management (Bressers et al., 2016; Edelenbos and Teisman, 2011), which is why there is a growing demand for more unified and collaborative management approaches (Pulwarty and Sivakumar, 2014; UNDRR, 2021). This is what the Brazilian Drought Monitor succeeds to do. As previously mentioned in this study (Section 2.3), even though the monthly drought severity map relies on broad and non-contextual indices, its function is more as a collaborative tool through the generated monthly discussions on localised drought conditions which ultimately improves institutional and operational capacities to respond to a drought event (Cavalcante De Souza Cabral et al., 2023).

Therefore, a drought impact index that is both localised and replicable is challenging, if not unachievable. This is due to the inherent challenge of "plurality" in scales and perspectives. There is no "best" combination of scale, level, or perspective for drought monitoring because of the complexity and varied impacts of droughts across different scales and stakeholders. The monitoring gap arises from this imbalance between 'broad and easy' monitoring and capturing the local context. It results from the necessity to select specific scales, levels, and variables due to the impossibility of encompassing all relevant perspectives and scales in monitoring. However, what might help bridge this monitoring gap is a focus on monitoring systems' resilience through non-extreme events, and stakeholder consultations, as we discuss below.

### 4.3    Practical implications and recommendations for monitoring of drought and drought impacts

While our study identifies the mismatches and blindspots in existing drought monitoring indices, it does not provide alternative indicators that could better address these monitoring challenges. In that sense, our work provides an analytical overview. Our research introduces a methodology for evaluating the suitability of existing indices for monitoring drought impacts on specific systems, scales, and levels.

As this study and the identified monitoring challenges are based on comparing two datasets, official and based on interviews, one notable limitation lies in the quality of such data. While we have frequently pointed out inconsistencies and shortcomings in the official data, we have also built our argument about monitoring challenges on that same data and its quality. However, this does not undermine our study's findings as this official data, with its inconsistencies and shortcomings, is precisely what decision-makers have to work with. The interview process is also subject to several forms of bias. These include positive memory bias (Adler and Pansky, 2020), where participants might emphasise positive memories over negative ones; memory bias (Grant et al., 2020), where current circumstances can influence past recollections; selection bias (Catalogue of Bias Collaboration, 2017), where interviewees may not fully represent the community; social desirability bias (Bergen and Labonté, 2020), where respondents might give answers they think are expected; and observer bias (Mahtani et al., 2018), where the interviewer could inadvertently influence responses. While these biases are inherent to the interview process and the setup of interviews can vary, we find that the overall trends identified are consistent with those observed in other communities, supporting the robustness of our findings despite these limitations.

These limitations serve as a blueprint for future research and improvements in drought monitoring. We advocate for the continuous and official monitoring of drought impact data by technical extension officers, whether agricultural or social, at the local municipality level. As we will develop later in the text, drawing from existing initiatives (Walker et al., 2024), such continuous monitoring would allow for a more accurate and reliable assessment of drought impacts, thereby improving the quality of drought interventions.

To date, there is no drought impact index that covers both physical and human drivers. Notable initiatives include the Water Poverty Index (WPI, Sullivan, 2002), which gauges 'water poverty' across scales but faces challenges of plurality (Sullivan et al., 2006). The recently introduced Days to Day Zero (DDZ) Index (Lankford et al., 2023) assesses the resilience of irrigated agriculture in semi-arid regions. The DDZ, although tailored for irrigation, underscores the need to also monitor non-extreme events and actions with both the WPI and DDZ tracking the escalation towards extremes rather than just the extremes themselves.

Monitoring non-extreme drought events can prompt anticipatory measures. By tracking these events, drought managers can begin to implement medium- and long-term strategies, ensuring they are better prepared when a severe drought does occur. Currently, this proactive approach is hindered by drought monitoring systems and official data, which focus on extreme events. They often detect an anomaly or a deviation from the average when corrective action is already more challenging, as the impacts already occurred. This need is highlighted in a recent study by Walker et al. (2024) also in the Brazilian semi-arid region. Their analysis of a drought impacts monitoring dataset from Ceará, showed that impacts still occur but are often normalised during mild or non-drought periods. The main drivers of these impacts were either non-extreme hydrometeorological conditions or socio-technical vulnerabilities.

In Walker et al. (2024), monitoring non-extreme drought impacts is delegated to agricultural technicians within the municipality, possessing rich local knowledge from past drought experiences and from operating in the communities within the municipality on the daily basis of their work outside of the monitoring. Though the reporting is at the municipal level, the nuances regarding how and why different communities are affected by drought in various ways can still be discerned, provided the technicians report it. This type of monitoring is a good compromise between what is logistically feasible in terms of monitoring and capturing the local nuances of (resilience to) drought impacts before they escalate to extreme levels, thereby helping bridge the monitoring gap.

Finally, it is important to note that another significant factor in the monitoring gap is that, even when human drivers of resilience to drought impacts are investigated, the challenge remains of how to integrate them into drought monitoring or early warning systems which are currently predominantly based on hydro-climatic drivers. Many human drivers of resilience and vulnerability to drought impacts are assessed qualitatively, as shown in this study (e.g. adherence to programs, diversification of the water, food, or livelihood system), or the Brazilian drought monitoring impact study (Walker et al., 2024, e.g.: high costs of energy, planting in low-lying areas) . Current drought monitoring systems often have a strict framework that does not easily accommodate qualitative data. Yet, qualitative observations play a pivotal role in local decision-making at household and community levels, which can have ripple effects at higher spatial levels or further in time (Kchouk et al., 2023a; Ribeiro Neto et al., 2023b). Therefore, an important challenge for drought monitoring lies in developing frameworks that allow the integration of such crucial qualitative data.

## 5    Conclusion

We developed a Monitoring Efficacy Matrix (MEM) to assess how well official data relevant to drought impacts align with community-level drought experiences, especially regarding impacts on water, food, and livelihood systems. By applying the MEM to the case of the rural community of Olho d'Água in Northeast Brazil, we identified monitoring challenges, consisting of mismatches and blindspots. At the community level, mismatches were caused by discrepancies between broad-scale data and specific local conditions, such as using municipal-level livestock and honey production data for small-scale farming, and drought data time-resolution not aligning with drought impacts duration or lag time. Blindspots emerged from important components of the indices not being accounted for, such as small reservoirs water volume, or from entirely missing the community's evolving resilience factors, such as irrigation and alternative crops. Our findings reveal that as the community's livelihood, food, and water systems diversified and became more resilient, the efficacy of drought monitoring decreased.

These mismatches and blindspots stem from the plurality of spatial and temporal levels pertinent to drought actors and impacts, as well as actions and strategies that determine a system's resilience to drought impacts. Given the challenge of considering all relevant scales and perspectives, drought monitoring often standardises or selects specific scales, levels, and variables to monitor. This approach, while aiming for simplification in drought governance and management, creates a monitoring gap by favouring 'broad and easy' monitoring at the cost of losing the local nuances of drought impacts.

A first step to bridge this drought monitoring gap is focusing on tracking systems' resilience by continuously monitoring non-extreme events and delegating this task to municipal technical extension officers. This type of monitoring offers a better balance between logistical feasibility at the municipality level and capturing local nuances of resilience to drought impacts at the community level. A second step, towards fully addressing this monitoring gap, would still require adaptations in drought monitoring and early warning systems, as current frameworks do not accommodate the integration of the qualitative nature of data associated with human drivers to drought impacts.

*Data availability*. Authors do not have permission to share the content of the interviews. However, a detailed narrative is provided. Data are available in the 4tu.ResearchData platform. The DOI and link of access is https://doi.org/10.4121/6edb96df-569e-41e8-9e6c-ba0a324c4729.v1

*Author contributions*. SK has designed and conducted the research in collaboration with LC, GRN, RG, and WJS. The research was supervised by LAM, DWW and PRvO. PRvO supervised and acquired financial support for the project leading to this publication.

*Competing interest*. The contact author has declared that neither they nor their co-authors have any competing interests,

*Acknowledgements*. We thank Petra Hellegers and Hela Gasmi who reviewed and provided helpful comment on earlier drafts of the manuscript. We thank the reviewers whose comments and suggestions helped improve and clarify this paper.

*Financial support*. This research has been supported by the Dutch Research Council (NWO) and the Interdisciplinary Research and Education Fund (INREF) of Wageningen University, the Netherlands (grant no. W07.30318.016).

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
