# Peer review of "Mind the Gap: Misalignment Between Drought Monitoring and Community Realities"

_EGUsphere, 2023_

## Author Response (AR1)

**REVIEWER 1**

Dear reviewer,

Thank you very much for your detailed, positive, and constructive review, which will certainly improve the quality of our manuscript.

The revised version of our manuscript will incorporate the majority of the suggestions. Responses to the points raised (normal text) can be found below (**in bold**)

On behalf of all co-authors,

Sarra Kchouk

2.1 Writing style

The paper is generally well written. Especially the discussion and conclusions are interesting to read. However, some parts are a bit lengthy and repetitive (addressed in the list of suggestions below). During the revision of the manuscript, please make sure to use a concise language as well as precise definitions and descriptions.

>>**Thank you for your positive evaluation and for identifying the repetitive sections. We will address these issues in the revision.**

2.2 IPCC risk concept

To understand the methods straightaway, it would be beneficial to apply the widely acknowledged IPCC risk concept throughout the manuscript (IPCC, 2014). According to this concept, drought risk (and impact as the manifested risk) is a function of drought hazard, exposure, and vulnerability. All these three components influence drought risk and impact. In a drought monitoring, it is crucial to first define the targeted risk system (the risk of what for whom) (which is basically your Table 4) and then to identify sector-specific indicators of drought hazard, exposure, and vulnerability that fit to the targeted risk (Hagenlocher et al, 2019; Meza et al, 2020 (with the latter reference already cited in the paper). By using these definitions, some redundant or unclear definitions can be avoided throughout the text.

Specifically, the conventional or physical indicators are drought hazard indicators, and the official data are impact data. Findings from the interviews represent qualitative impact data. These definitions should be used consistently in the manuscript.

The three different periods in Table 4 are distinguished based on the different levels of vulnerability due to the change from rainfed to irrigated agriculture, the construction of small reservoirs and other water storages, and the diversification of crops. I understand that this is addressed within the text, but it should be highlighted at the beginning when introducing Table 4. It should become clear that Table 4 presents the targeted risk systems (the risk of what for whom) and that the columns are characterized by different levels of vulnerability.

>>**Thank you for suggesting the application of the IPCC risk framework. While we acknowledge that this is a well-known and accepted approach, we believe it may align partially but not fully with the specific objectives and nuances of our study.**

>>**We do not start from the assumption that conventionally monitored data necessarily represents hazard data. The categorization of hazard, exposure, vulnerability, and impacts, while useful in some cases, does not always align perfectly with optimal drought monitoring practices. We find that these categories can overlap. For instance, monitored agricultural data can be seen as impact data or exposure data, depending on the context. For example, in the context of water security,**

'reservoir level' could reasonably fit into exposure, vulnerability, or impact categories. This flexibility in categorisation reflects the overlapping nature of indicators, which can fit into multiple categories depending on the context.

Adopting the IPCC framework would require us to make rigid distinctions that might not capture the complexities of our data. This could lead to increased length and redundancy in the manuscript, as we would need to justify the categorisation of each indicator. More importantly, this could detract from our primary focus, which is to highlight the mismatch between (i) what is monitored and available for decision-makers, and (ii) community-level experiences of drought and its impacts.

Most importantly, even if we follow the IPCC framework, we believe the primary shortfall of DEWS, and therefore our premise, is twofold: they exclude the human aspects of drought, such as the anthropogenic drivers (i.e., the anthropogenic aspects of exposure and vulnerability) and the impacts on human populations. Additionally, they overlook the necessary contextual aspects of the human dimensions of drought, which we believe transcend the explanations that the IPCC framework can provide.

We will clarify our objective in the manuscript to prevent any confusion. However, we believe that using the IPCC framework to define drought and refine our research gap, as explained above, is extremely useful. Therefore, as recommended by Reviewer 2, we have amended our introduction accordingly.

L1: More and more studies highlight the human influence on droughts, demonstrating that drought results from both natural and anthropogenic drivers (Aghakouchak et al., 2021; Van Loon et al., 2016; Walker et al., 2022; Di Baldassarre et al., 2018) and affects the hydrological cycle and human populations (Savelli et al., 2021; Ribeiro Neto et al., 2022; Kchouk et al., 2023a). Despite this recognition, limited progress has been made in incorporating this knowledge into drought monitoring and early warning systems (DEWSs). DEWSs still predominantly rely on hydro-climatic indices, focusing only on some aspects of drought risk. According to the IPCC framework, drought risk is the interaction of hazard, exposure, and vulnerability (UNDRR, 2021; Carrão et al., 2016). Drought hazard refers to the failure of the system maintaining the hydrological balance, which can include e.g. reduced rainfall over a certain period, inadequate timing or ineffectiveness of precipitation, or a negative water balance due to increased atmospheric water demand from high temperatures or strong winds (UNDRR, 2021). Exposure involves the elements within a system— such as assets, infrastructure, species, ecosystems, and people—that could be adversely affected by the drought hazard (UNDRR, 2024). Vulnerability encompasses the physical, social, economic, and environmental factors that increase the susceptibility of these elements to drought impacts (IPCC, 2014; UNDRR, 2021).

DEWSs often lack indices that account for the human dimensions of drought, specifically: (i) human influences on drought risk and (ii) the impacts of drought on human populations. One reason for the focus on hydro-climatic factors in drought monitoring is the intertwining of physical and anthropogenic aspects within drought risk, making it challenging to quantify and attribute anthropogenic contributions precisely (Aghakouchak et al., 2021). Anthropogenic drivers add complexity and variability, as they are dynamic and non-static, which complicates their integration into monitoring systems (Kchouk et al., 2023a). Similarly, monitoring drought impacts is difficult because they are non-structural, difficult to quantify or monetise, and can be direct or indirect (Bachmair et al., 2016; Kchouk et al., 2023a; Logar and Van Den Bergh, 2013; Wilhite et al., 2007).

The lack of accounting for human drivers of droughts and their impacts in DEWSs results in what we call a "drought-monitoring gap": a gap between the monitored data and the drought conditions experienced by human populations. This oversight hinders DEWS from fully achieving their goals of facilitating proactive, well-informed decision-making and providing vulnerable groups with

timely, reliable data (Pulwarty and Verdin, 2013; UNDRR, 2021). The local context's specificity adds to this challenge, as understanding and monitoring these human dimensions are complex and context-dependent (UNDRR, 2021). This gap complicates drought managers' ability to make informed decisions and take appropriate action, especially when the anthropogenic drivers of drought risks and impacts remain unknown outside the affected area. Although there are promising drought-impact-monitoring initiatives (EM-DAT; EDII, 2023; Smith et al., 2024), these efforts are still in the early stages and have mainly enhanced our understanding of drought risk and its impacts (Lackstrom et al., 2022; Stephan et al., 2021; Tijdeman et al., 2022). While this progress is an important step forward, further work is needed to build on this foundation—particularly by incorporating people's lived experiences of drought impacts—to develop actionable strategies that address underlying vulnerabilities and build resilience to future droughts.

2.3 Drought monitoring in the case study area

Since the authors make suggestions on how to improve drought monitoring / management in the case study area, it would be informative to read how drought monitoring, management, and reporting is currently carried out in this region. For instance, which operational drought definition is in place? From the conclusions, I derive that drought response measures are carried out in the case of severe droughts (?). Also, in Table 5 you mention the threshold -1 for SPI. Is this the existing operational drought definition?

**>>Thank you for these suggestions.**

**SPI <-1 indicates a rainfall deficit. We will amend L130 as follows:**
**L130: We extracted data from various international and Brazilian databases (Table 2), covering climatology, reservoir storage, and agricultural production. We used the rainfall data to calculate different Standardised Precipitation Indices (SPIs), each serving a distinct purpose: SPI-3 indicates short-term soil moisture conditions, which are relevant for crops; SPI-6 provides a mid-term perspective, affecting agriculture and offering early signs of changes in water storage; SPI-12 monitors long-term trends in water storage and streamflow (WMO, 2012). An SPI of -1 or lower indicates conditions indicates a rainfall deficit, signifying conditions drier than normal across the relevant time scale (EDO, 2020).**

**Copernicus European Drought Observatory (EDO): Standardized Precipitation Index (SPI) - EDO Indicator Factsheet. European Commission Joint Research Centre. Available at: https://drought.emergency.copernicus.eu/data/factsheets/factsheet_spi.pdf (last access: 12 August 2024), 2020.**

**Please note that from L131 onwards, we detail how drought monitoring is operationally conducted in the area. We will emphasise that the same approach applies to the studied community, and instead of focusing on a single 'operational' definition of drought—such as agricultural or hydrological—drought monitoring integrates indices from different categories. We will amend the paragraph as follows:**

**L131: Since 2016, the Brazilian Drought Monitor has produced a monthly map of drought conditions, based on drought indices at multiple computation periods from one to twenty-four months. Instead of focusing on a single operational definition of drought—such as agricultural or hydrological—the Drought Monitor integrates indices from different categories to provide a comprehensive understanding of drought conditions. These indices are SPI, Standardsed Precipitation and Evapotranspiration Indices (SPEIs), Standardized Runoff and Dry Spell Indicators, and remote-sensing indices, and are validated by regional offices that consider ground**

**observations from networks of observers (De Nys et al., 2016; Walker et al., 2024). The same approach applies to the studied community of Olho d'Água, located in the municipality of Piquet Carneiro, which is included in the maps produced by the Brazilian Drought Monitor. The Drought Monitor categorises conditions starting from 'no drought' to 'weak drought', which indicate the beginning or end of dry conditions. Categories of 'moderate', 'severe', 'extreme', and culminating in 'exceptional drought', indicate widespread losses in crops and pastures and water shortage at an emergency level.**

As per my understanding, the examined indicators are the only ones available for the study region and (partially?) used by the relevant institutions to trigger drought response measures? Some explanations in the text (e.g., "we chose to examine time series…", L 75) could also mean that the authors made a selection from a set of available indicators. Then, it would be important to explain why other available indicators were not considered.

**>>Thank you for your suggestion. This is a fair point. Some indices are indeed used by relevant institutions, while others can be used by a range of individuals, from policymakers to small farmers, to inform their decisions. Therefore, we use the general term 'decision-maker.' We selected indices based on the most accessible data from the most widely accessed portal in Brazil, according to our experience, and also the most commonly used indices in drought monitoring. Please note that this is detailed and explained from L72 onwards:**

**Step 2: We examined conventional drought indices and officially monitored data relevant to drought impacts that could characterize drought conditions in the focus area. "Conventional indices" refer to commonly used metrics to quantify and characterize drought conditions. These include time series characterizing rainfall and meteorological drought (SPIs, Brazilian Drought Monitor Map), agriculture (cropped and harvested areas, crop yields, agricultural output), and hydrology (reservoir volumes and water surface area). These drought indices and official data are among the most widely used and agreed-upon tools to monitor and characterise drought severity (Bachmair et al., 2016; Kchouk et al., 2022), and they align with the impacts on livelihood, food, and water security that we aimed to explore. These datasets have specific spatial and temporal monitoring levels, which are not necessarily homogeneous across the different datasets, nor aligned with the levels at which impacts are experienced by populations. Specific information about the data series and data collection is provided in Section 2.3.2.4**

Specific comments in the text

L 13 and L 29: I suggest replacing the term "physical indices" by "drought hazard indices" or "hydro-climatic indices". Physical indicators can also be indicators of drought exposure, for example.

**>>Thank you for the suggestion. We will use 'hydro-climatic indices'.**

L 40: Please add: drought hazard indices

**>>Thank you for the suggestion. For coherence, we will use hydro-climatic indices.**

L 40: Could you rephrase the last part of the sentence: "are unknown…" (unknown to drought managers?)

**>>The text has been amended as follows:**
**This gap complicates drought managers' ability to make informed decisions and take appropriate action, especially when the anthropogenic drivers of drought risks and impacts remain unknown outside the affected area**

L 72: Please specify: "the available conventional drought hazard indices

**>>Thank you for the remark, but as detailed above, we explain what is meant by 'conventional indices'. As noted, not all the indices we chose can be considered purely as drought hazard indices—some represent impacts, while others can be viewed simultaneously as indicators of exposure and vulnerability. Furthermore, we specify immediately afterward what these conventional drought indices entail, coining this term to maintain conciseness throughout the manuscript. This approach allows us to avoid redundancy by not repeatedly stating "the hydro-climatic indices available online…" each time.**

L 73: Please delete this sentence, since it is too unspecific: "Conventional indices"…

**>>Please see the comment above. We do not believe this is too unspecific, as we clarify what is meant by 'conventional indices' immediately after introducing the term. Furthermore, at L80, we refer to Section 2.3, where we provide detailed information on the specific indices we examine.**

L 74: To be more concise, I suggest starting this sentence with: "These include" instead of "To achieve this, we chose to examine time series characterizing". Also, "we chose to examine" suggests that the indicators were selected from a larger number of available indicators. From my understanding, these are the only indicators that are available and applied for the study area?

**>>Thank you for the suggestion, we will amend L74 to start with 'these include'.**

Also, I would suggest to first list the examined hazard indicators (SPI, Brazilian Drought Monitor, reservoir volume and water surface area), and second the impact data (and describe them as hazard and impact indicators).

**>>Thank you for the suggestion. As mentioned earlier, our approach does not opposed hazard and impact indices in drought monitoring. Additionally, we believe it is best to maintain the original structure, as it aligns with the structure of the results subsections.**

L 75: Please shortly describe the Brazilian Drought Monitor here. Is it a combination of indicators and expert information like the US Drought Monitor?

**>>Thank you for your suggestion. Please note that this is specified L131 and addressed in a previous comment.**

L 77: and are also related to the impacts (instead of "fit the impacts"). You show later that they do not always fit.

**>>Thank you for the suggestion. Indeed, we will amend accordingly.**

L 81: "We compare the data" instead of "findings".

**>>The sentence will be amended accordingly.**

L 82: Please delete the sentence "This framework…" as it is redundant.

**>>The sentence will be deleted.**

L 90: productive activities = irrigation? Please clarify here.

**>>We will clarify using the terms: Income-generating and assets-building activities**

L 93: "smallholder farmers reliant on rainfed agriculture". Table 4 lists only irrigated agriculture after 2003. If rainfed agriculture is still relevant after 2003, then it should also appear in Table 4 after 2003.

**>>The sentence at L93 refers to the entire semi-arid region during the latest multi-annual drought, as mentioned earlier in the text :**
**L91: The Brazilian state of Ceará, located in the semi-arid region known as Sertão, has faced consistent drought challenges (UNDDR, 2021). The latest multiannual drought (2012 – 2019), noted for its intensity, deeply affected the region's agriculture. Most impacted were smallholder farmers reliant on rainfed agriculture, who experienced significant crop losses and economic setbacks (Brito et al., 2018; Pontes Filho et al., 2020) as well as compromised water availability and quality (Eakin et al., 2014).**

L 124: Please define "semi-structured" and "unstructured".

**We will add to L115: The interviews were both unstructured, allowing for open-ended, flexible discussions without predetermined questions, and semi-structured, guided by a set of predefined questions while the rest of the questions are not planned in advance, permitting exploration of other topics in depth (Brinkmann, 2014)**

**Brinkmann, S.: Unstructured and semi-structured interviewing, The Oxford handbook of qualitative research, 2, 277-299, 2014.**

L 142: Please shorten the first paragraph of Section 2.4 and use a more concise language to avoid redundancies throughout this section. I think the method would be easier to understand if you mention once at the beginning that it can be applied in the context of different phenomena, and then describe it for drought only (using only examples that are related to drought).

**>>Thank you for the suggestion. The paragraph has been amended as follow:**

**L142: Monitoring efficacy refers to how effectively a system detects, tracks, and reports on specific parameters or events. In the context of drought, this includes the accuracy, reliability, timeliness, and comprehensiveness of drought indices used by monitoring systems. Drought-monitoring efficacy, therefore, relates to how well these systems detect and report drought conditions, including severity and impacts. Challenges arise when systems fail to capture these conditions accurately, reliably, or comprehensively. We propose a drought-monitoring efficacy matrix to identify and address these challenges.**

L 159: Please delete this sentence as it is redundant. Also, it suggests that the indicators examined are only a selection by the authors and do not represent the only indicators that are available and applied in the study area (see also comment L 74). Please avoid the term "fit" in this context as you show with the MEM that some indicators do not fit the targeted risk.

**>>L159 has been amended as follow:**

**These indices were selected because they align with the impacts we sought to explore, specifically those related to livelihood, food, and water security at both household and community levels.**

L 184-200: I would suggest deleting these two paragraphs as they present many findings that you describe in the results section. From my point of view, the general definition of mismatches and blindspots that you provide is sufficient at this point.

**>>Thank you for this suggestion. We agree that these sections might seem redundant, particularly due to the examples chosen. However, we believe they are essential for understanding the concept of mismatches and blindspots. Additionally, they are relevant for readers who may want to focus on the methods section without reading the results, as is sometimes the case.**

L 211: Table 4: Please mention in the table caption, why three different periods are distinguished (different levels of vulnerability).

**>>Thank you very much for these suggestions. However, as the table's caption indicates, it is not solely about vulnerability. Following the framework suggested by the reviewer, it also addresses exposures, such as activities that generate income or build assets. Focusing only on the evolution of vulnerability would be reductionist. Moreover, vulnerability is dynamic, and discussing its fluctuations would require a more detailed explanation, which is more appropriate for the discussion section. The purpose of this table is to provide an overview of income strategies, or as previously highlighted by the reviewer, to present the data rather than the findings.**

L 257: I would suggest using a header that is more informative regarding what you assess or discuss in this section. Furthermore, I would use more specific terms for the indicators (monitored hazard indicators and official drought impact data). I would avoid the term "global data" as it has not been mentioned before and is too unspecific here.

**Thank you for the suggestion. We will remove the term 'global data'. The meaning of conventional and official data has already been defined and established in the methods section. Using it here is consistent with what was outlined in the methods.**

L 260: Suggestion: "agricultural impact data" instead of "agricultural data".

**Thank you for your suggestion. Agricultural 'impact' data would refer to deviations from the norm, which is not the case here. We are using agricultural production time series and analysing their evolution over time.**

L 269: Please clarify if these are the only SPI variants that are applied by the relevant institutions. Otherwise, please explain why you chose the averaging periods 3, 6, and 12 for SPI.

Thank you for the suggestion. We will amend the text as follows:

**L128: We used the rainfall data to calculate different SPIs, with each indicating a different purpose: SPI-3 indicates short-term soil moisture, relevant for crops; SPI-6 provides a mid-term view, affecting agriculture and early 130 signs of water storage changes; SPI-12 monitors long-term trends in water storage and streamflow (WMO, 2012).**

**L131: Since 2016, the Brazilian Drought Monitor has produced a monthly map of drought conditions, based on drought indices at multiple computation periods from one to twenty-four months. These indices are SPI, SPEI, Standardized Runoff and Dry Spell Indicators, and remote-sensing indices, and are validated by regional offices that consider ground observations from networks of observers (De Nys et al., 2016; Walker et al., 2024). The Drought Monitor categorises conditions starting from 'no drought' to 'weak drought', which indicate the beginning or end of dry conditions. Categories of 'moderate', 'severe', 'extreme', and culminating in 'exceptional drought', indicate widespread losses in crops and pastures and water shortage at an emergency level. We retrieved data relevant to large and strategic, and small and non-strategic reservoirs. Finally, we obtained agricultural data encompassing the relevant crops in the community of Olho d'Água.**

L 282: Please describe in the caption of Figure 2 the severity classes according to the Brazilian Drought Monitor. Please also mention that the monitoring of the Sao José dam began in 2004.

**>>Thank you for the suggestion. The caption will be amended as follows:**

**Figure 2: (a) Annual crop quantity produced in tons in the municipality of Piquet Carneiro from 1974 to 2019 (lines). Daily evolution of the percentage of the total volume of the Sao José II dam, which monitoring started in 2004 and is the only monitored dam in the municipality of Piquet Carneiro (blue shaded area); (b) Annual equal cropped and harvested area in Piquet Carneiro from 1981 to 2021 (lines). The columns represent the annual crop yield per hectare. (Source: IBGE, PAM); (c) Annual equal cropped and harvested area of banana in Piquet Carneiro from 1981 to 2021 (lines). The columns represent the annual crop yield per hectare of banana. (Source: IBGE, PAM); (d) Annual livestock population and production in Piquet Carneiro with the number of cows' head (black line), the annual milk production (in thousands of litres, red line), and the annual honey production (in kg, orange columns) (Sources, IBGE and Conab). Below are the colour bars of the monthly values of the SPIs 3, 6, and 12. At the bottom are the monthly percentages of the municipality under different categories of drought severity, from July 2014 to November 2022 (white: no drought, yellow: weak drought, light orange: moderate drought, dark orange: severe drought, red: extreme drought, brown: exceptional drought ; Sources: Cogerh/Funceme, Brazilian Drought Monitor, CHIRPS).**

L 331: Table 5:

Table 5A, header: "SPI values below -1": Is this the operational drought definition triggering drought management responses? Please clarify at some point.

**>>Thank you for the suggestion. We previously detailed in the text that this value indicates drier conditions, as mentioned in an earlier comment.**

Table 5A: Please distinguish hazard indicators and impact indicators in the header.

**>>As mentioned before, the comparison focuses on conventional and officially monitored drought risk and impact-related data versus the lived experiences of the communities.**

Table 5A: Since the population did not depend on reservoir storage before 2003, I think it is incorrect to identify the missing reservoir data (water area and reservoir level) as a blindspot. I think this should be replaced by N/A.

**>>Fair point, it will be replaced by N/A.**

Table 5A: The column for Crop yield/ha is left blank. Is this N/A?

**>>Thank you. Indeed, it is N/A**

Table 5A: M1 for „Quantity produced" and „Area harvested", while the description of M1 only refers to livestock. Is M1 correct here?

**>>Thank you very much for pointing this out. Indeed, M1 is incorrect. We will replace it with M0: mismatch in terms of whether larger-scale cotton data can be accurately applied to the community.**

Table 5B: I cannot follow the explanation for the match for SPI and water security. Since people still used water from shallow and deeper wells and newly installed cisterns, I would suggest using the same explanation as before in Table 5A, which seems still valid here for shallow groundwater and cisterns.

**>>The reviewer is correct, and this is explained in the corresponding case of SPI and water security. Surface water sources are highly reactive to rainfall and drought. However, during this period, no prolonged droughts were captured by the SPI (as detailed in L351), so the water sources were not affected by a rainfall anomaly. This observation aligns with the community's experience, indicating a match.**

Table 5D: Description „Blindspots": Blindspots were also attributed to the SPI column (= drought hazard), so they are not only related to reslilience and vulnerability.

**>>This is a very good point. We will use the term 'risk'  instead (as it encompasses vulnerability, hazard, and exposures) and amend it as follows:**

**Blindspots: not all the elements of resilience or risk of the systems to drought impacts are monitored**

L 340, 341: As outlined for Table 5A, I do not think that this is a blindspot, since there was only rainfed agriculture in place before 2003.

**>> Thank you for this suggestion. We agree that not monitoring the reservoir level is not a blindspot, as the reservoir had not yet been constructed before 2003. However, hydrological data for water areas smaller than 0.5 ha are relevant because the community relied on shallow wells for water security prior to 2003. Therefore, the lack of such data does represent a blindspot. This will be clarified in Line 340 as follows:**

**During this time, there are no available hydrological data for areas smaller than 0.5 ha, creating a blindspot given the community's reliance on shallow wells for water security before 2003.**

L 345: If irrigation was relevant for cotton before 2003, it should have a separate row in Table 5A. Otherwise, please clarify that irrigation did not play a role before 2003 based on the interviews.

**Thank you for this very good point. L345 will be amended as follows:**
**The stable trend in cotton during droughts also suggests irrigation, in contrast to the community's water scarcity experiences reporting full loss and stating that they were not irrigating prior to 2003 (Mismatch 12).**

L 346: Mismatch 12 is not included in Table 5.

**Thank you for this observation. It is indeed a typo on our part, and it will be corrected at L345 to refer to Blindspot 11, which pertains to blindspots related to irrigation practices. The trend of the monitored variable suggests irrigation, for which there is a blindspot, as we do not have any official monitoring data in that regard.**

L 368: What type of capacities? "Eroded in the absence of…" is also not clear to me.

**>>Thank you for this suggestion. L368 will be amended as follows:**

**However, during the prior decade, the community developed or benefitted from resilience mechanisms for their water, food, and livelihood systems. These mechanisms also remained robust as they were not weakened by any severe droughts during that period. Therefore, despite the 2012-2019 drought affected the livelihood, food, and water systems, they were not as severely impacted as they were before 2003, because of alternative governmental measures like Bolsa Família, Garantia Safra, PAA, PNAE, and water trucks.**

 L 376: Mismatch 7 instead of 17. Why is this a case of wrong spatial or temporal scale?

**>>Thank you. The typo L376 will be corrected to M7. Mismatch 7 in Table 5D will be amended as follows:**

**M7: Data stable or upward trend suggests irrigation, which was not the case as the community declared water insecurity and the impossibility of irrigating. This suggests that the monitoring might have been focusing on a single farm irrigating, thereby challenging its generalisability across the entire municipality.**

L 395: Please add: conventional drought hazard indicators and official impact data

**>>Thank you for this suggestion. As we have detailed previously, we will keep the terms "conventional drought indices" and "official data" because we have explained their meaning in the Methods section. We want to maintain coherence and better align with the idea we are trying to convey.**

L 397: Please add: between spatial and temporal levels of monitoring

**>>Thank you. L397 will be amended as suggested.**

L 398: Suggestion: of all elements of drought hazard, vulnerability and exposure that are relevant for the focus system.

**>>Thank you. We agree will amend the sentence as suggested.**

L 398: "that can be impacted by…": It is not clear if this refers to the indicators or the system. I'd rather delete this sub-sentence.

**>>Thank you. We will delete it**.

L 405: Sentence is redundant (see L 403.)

**>>We will amend L405 as follows: During the first period, when the community was still largely dependent on rainfall, the monitoring aligned reasonably well with experienced drought impacts, although it remained incomplete.**

L 406: Suggestion: "In the following periods, as the community diversified its livelihood, food, and water sources, the monitoring gap also increased. This indicates that as the systems became more complex and resilient, such that conventional indicators indices and data became […]"

**>>Thank you for this suggestion. The text will be amended as suggested.**

L 471: Suggestion: two impact datasets

**>>As mentioned earlier in authors' response, while the community's experiences can be considered an impact dataset, the same cannot be said for the 'conventional and official dataset,' which mixes all components of drought risks and impacts without straightforward categorisation.**

L 506: hydro-climatic rather than physical drivers

**>>We agree and will amend the sentence as suggested.**

L 514: Suggestion: "how well official monitoring data comprising drought hazard and impact indicators relevant to drought impacts align with […]"

**>>Thank you for this suggestion but as mentioned earlier, the "conventional and official dataset' mixes all components of drought risks and impacts.**

L 528: Suggestion: by continuously monitoring drought impacts of non-extreme events

**>>Thank you for this suggestion, but we do mean monitoring all related data for non-extreme events, including continuously monitoring drivers and impacts data.**

3. Technical corrections

L 44: Meza et al., Carrao et al ,and Haile et al. not in the list of references.

**>>Thank you very much. We will add it to the references.**

L 269: "belowdepicts": Please add blank space.

**>>Thank you very much. We will amend it.**

L 291: Given the low resolution of Figure 3, it would be better to replace the red dot by, e.g., a square to ensure that readers with colour vision deficiencies can distinguish the locations.

**>>Thank you very much. The resolution will be improved, the red dot changed and the size of the figure extended so it is easier to distinguish the two different locations.**

L 304: Figure 2a instead of 3a

**>>Thank you very much. The text will be amended.**

L 353: Please change: […] the resilience […] has increased.

**>>Thank you. It will be amended as: because the resilience of the systems to drought has increased**

L 379: Figure 2a

**>>Thank you very much. The text will be amended.**

L 384: "The same mismatch…": Please check the grammar of this sentence.

**>>Thank you. The sentence will be amended as:**

**The same mismatch is evident in the livestock data, where a trend showing 12,000 to 20,000 cows is too broad to reflect the local average of five cows per household (Mismatch 1).**

Dear Dr Natasha Pauli

Thank you for your thoughtful, constructive, and positive feedback and for taking the time to provide detailed suggestions, that will certainly increase the quality of our manuscript.

We noted the suggested main areas for refinement are broken down into specific comments that are detailed and answered to, below. The revised version of our manuscript will incorporate the majority of the suggestions. Responses to the points raised (normal text) can be found below (**in bold**)

On behalf of all co-authors,

Sarra Kchouk

This paper presents an interesting mixed methods approach to understanding community needs for drought monitoring, using a case study approach combining interviews and data extraction from various portals. The method presented is novel and will likely be of interest to researchers and practitioners in the meteorology and agricultural communities.

The authors' assertions of the need to take into account physical and anthropogenic drivers of drought and water insecurity is well made – although, more clarity could be provided on what are the 'human drivers' of drought.

The main areas suggested for refinement in the manuscript are as follows:

An expanded definition of what is meant by 'drought' should be present early on the manuscript. There are multiple, context-specific interpretations of what is meant by the word 'drought'. In some countries, it might be years without rain, and in other places, it might be an unusually low rainfall growing season. As the subject of this paper, drought deserves a more thorough, critical exploration.

The origin and development of the Monitoring Efficacy Matrix is not clear. Was this developed based on other similar frameworks, or is it developed as part of this research – no citations or other examples of the framework are provided in the paper. Further explanation is required.

The section reporting on the narrative and interview results presents one 'story' of the case study location. Some discussion of topics where there was disagreement, or vague memories, or that were not brought up by all respondents is warranted.

No detailed justification is provided as to why the very small case study community was chosen as the location for this research. Why not another community in Pique Carneiro? At some point in the manuscript, a wider research project with more interviewees is mentioned - is this the reason why Olho d'Água was selected, as part of the other research? Some justification of the location is warranted.

Specific comments follow:

Line 34: 'These have largely served only to increase the understanding of drought' – something is missing in this sentence. It isn't clear what areas are still deficient in understanding.

**Thank you for the suggestion. The text will be amended as follows:**

**>> Although there are encouraging drought-impact-monitoring initiatives (EM-DAT; EDII, 2023; Smith et al., 2024), these efforts are still in the early stages and have primarily contributed to enhancing our understanding of drought and its impacts (Lackstrom et al., 2022; Stephan et al.,**

**2021; Tijdeman et al., 2022). While this is an important step forward, further work is needed to build on this foundation—especially by incorporating people's lived experiences of drought impacts—to develop actionable strategies or policies that address underlying vulnerabilities and build resilience to future droughts.**

Line 70 and onwards, Methodological Approach: It is suggested to write in paragraphs what the overall approach is, rather than bullet points and 'step 1, step 2', particularly because the bullet points are already quite long.

**>>Thank you for the suggestion. We will remove the bullet points and alignments, and present the steps in paragraph form.**

Line 75: SPIs are not expanded the first time they are mentioned.

**>>Thank you. SPIs will be expanded to Standardised Precipitation Indices.**

Line 80: No citation or information on the origin of the MEM is provided.

**>> Thank you for this suggestion however we would like to highlight that the origin of the MEM is provided mentioning that it is a novel approach, L80, but also in the introduction L60. Furthermore, in L80, we mention that the specificities of the MEM, which include its origins, are further detailed in Section 2.4. We will amend L60 to make it clearer that we developed this framework:**

**L83: We designed this framework to examine the alignment of a drought index with reported impacts.**

Line 88: Why Olho d'Água? Is this a very typical town? Is it atypical? What does it represent?

**>>Thank you very much for pointing that out. We will add the explanation L118:**

**While the fieldwork campaign provided valuable insights that led to further studies focusing on different communities (Walker et al., 2024; Ribeiro Neto et al., 2023; Kchouk et al., 2023; Cavalcante et al., 2024), we focused in this study on the community of Olho d'Água due to its atypical characteristics. Unlike other investigated communities, Olho d'Água has diversified means of water, food, and livelihood security, and, relatively speaking, did not experience a total collapse of these systems during or after the multi-annual drought of 2012 to 2019 (see Kchouk et al., 2023). Therefore, understanding the reasons behind this resilience and whether these factors are being adequately monitored became a key focus.**

Section 2.2: Please explain whether the terms 'monitored' and 'non-strategic' are incompatible. Can there be a 'monitored non-strategic' reservoir? Or is it that all strategic reservoirs are monitored, and all non-strategic reservoirs are unmonitored? The repeated use of these terms and the addition of 'officially' and 'informally' and 'formal' is rather confusing, as is the phrase 'officially unmonitored'. Suggest to first explain what is meant by 'strategic' and non-strategic reservoirs early in the section, and then describe the communities.

**>>Thank you for your suggestion. We have amended L99 as follows:**
**The distinction between what are colloquially referred to as strategic and non-strategic reservoirs is crucial for understanding the local context and the associated monitoring challenges. Strategic reservoirs are large public infrastructure projects that are systematically monitored by state water agencies, primarily serving urban populations. In contrast, non-strategic reservoirs are smaller reservoirs (under 1 million m³ (Rabelo et al., 2022)) that are typically constructed by rural populations to ensure their water access. While strategic reservoirs are always monitored, non-strategic reservoirs are typically unmonitored, though there may be exceptions, as they fall**

**outside the official state-planned reservoir grid. However, these non-strategic reservoirs remain informally strategic at the local level because the majority of rural communities depend on them for their income-generating activities. As these smaller reservoirs are locally built and managed, they elude the control, maintenance, and monitoring of official agencies.**

Line 116-118: The rest of the fieldwork campaign is not mentioned again – is it relevant? If it is relevant please include a citation to the other work(s).

**>>Thank you very much for this comment. We agree and addressed it in a previous response.**

Line 119: How long were the interviews (range in minutes)? Were notes taken during the interview? Was ethics clearance required?

**>>Thank you for your comment. We amended L119 as follows**

**All interviewees provided consent before being interviewed, in accordance with ethical clearance requirements. The interviews, which lasted between twenty minutes and three hours, were not recorded but were immediately transcribed to ensure the comfort of the interviewees who were predominantly smallholder. None of the solicited individuals declined to be interviewed.**

Table 1: Unclear what is meant by the column heading 'query category'. Is this rather 'question themes'?

**>> Thank you for your suggestion. We will replace the 'query category' by 'question themes'.**

Table 1: Mentions 'unstructured' and 'semi-structured' interviews. Why the distinction? – and also, suggest to explain why the authors chose these forms of interviews, and explain in particular what is meant by 'unstructured'. Unstructured can mean that it was a spontaneous conversation about whatever the interviewee wanted to converse about – but several 'query categories' are listed, which suggests that they may have been semi-structured, or unstructured but with a few broad topics in mind to start the discussion. In any case, it is suggested to provide some context/references on the methods that were chosen.

**>>Thank you very much for your suggestion. We agree that this specification is confusing in the table and redundant, as it has already been addressed in the previous paragraph. Consequently, we will remove this column from the table.**

Line 128: SPIs still not expanded.

**>>Thank you. As per one previous comment, SPI has been expanded L70.**

Line 131: SPEI not expanded.

**>> Thank you. We will expand SPEIs to Standardised Precipitation Evapotranspiration Indices.**

Line 142: CHIRPS not expanded.

**>>Thank you. CHIRPS will be expanded to Climate Hazards Group InfraRed Precipitation with Station data.**

Table 3: It is not clear what is supposed to go in the boxes of this matrix. Often such matrices (e.g. risk matrices) are filled with qualitative scores or categories such as high/medium/low risk etc. This looks less like a 'matrix' and more like a standard table or framework. Please provide some context as to how the matrix is actually supposed to be used/scored, for the benefit of others who may want to use or adapt this matrix. Unclear what is the difference between 'livelihood system' and 'food security' in a smallholder system (indeed in the later matrix items in these categories appear to overlap). Where did the three categories of 'livelihood', 'food security' and 'water security' come from? Suggest to provide a framework that serves as the basis for these categories (e.g. Line 161 – provide citations).

Thank you very much for these suggestions that will be included as follows:

[revised manuscript text omitted]

**World Bank: What is food security? Available at: https://www.worldbank.org/en/topic/agriculture/brief/food-security-update/what-is-food-security (last access: 12 August 2024), 2024.**

**UN Water: What is Water Security? Infographic: available at: https://www.unwater.org/publications/water-security-infographic/(last access: 12 August 2024), 2013.**

Line 190: 'Flash drought'? The meaning of 'flash flood' is widely understood but I am not sure if the same is true for flash drought – is it worth providing a definition of flash drought (perhaps earlier on in the MS when elaborating on the different contextual meanings of drought).

**Thank you very much for this remark. Referring back to the first suggested area for refinement, we have added earlier in the manuscript an elaboration on the different contextual meanings of drought.**

**L3: According to the IPCC framework, drought risk is the interaction of hazard, exposure, and vulnerability (UNDRR, 2021; Carrão et al., 2016). Drought hazard refers to the failure of the system maintaining the hydrological balance, which can include e.g. reduced rainfall over a certain period, inadequate timing or ineffectiveness of precipitation, or a negative water balance due to increased atmospheric water demand from high temperatures or strong winds (UNDRR, 2021). Exposure involves the elements within a system—such as assets, infrastructure, species, ecosystems, and people—that could be adversely affected by the drought hazard (UNDRR, 2024). Vulnerability encompasses the physical, social, economic, and environmental factors that increase the susceptibility of these elements to drought impacts (IPCC, 2014; UNDRR, 2021).**

**A definition of flash drought will also be added**

**L190: An example is the SPI-1, the shortest SPI, which sometimes overlooks impactful flash droughts; because it is based on monthly data, it cannot detect dry spells shorter than a month, and thus misses the rapidly developing and intensifying flash droughts that can occur within weeks or even days of precipitation deficits (Walker et al., 2023).**

Line 195: 'Count' should be something different like 'number'. The phrasing of this whole sentence is a little awkward and could be simplified.

**>>Thank you for the suggestion. L195 will be amended as follows:**
**For example, a blindspot can occur when small reservoirs, crucial to many communities' water systems, are only counted rather than having their volumes monitored.**

Table 4: Uses 'livelihood system' and 'food system' and 'water system' whereas previous tables used 'food security' and 'water security'.

**>>Thank you for this remark. The two tables have different purposes and captions. Table 3 is an empty MEM, using examples of food, water, and livelihood securities to assess whether they are impacted or monitored. In contrast, Table 4 is a value-neutral summarising table of the following sections, focusing on what composes the systems, not on whether they are secure or not. For**

**example, prior to 2003, the community was quite insecure, but the table describes what the water system was relying on rather than assessing its security.**

Line 222: Is it possible to define the policies that changed in 2003?

**>>Thank you for your suggestion. These policies are detailed L225 onwards.**

Line 247: Are there any census data available on population demographics of Pique Carneiro which would help illustrate these changes?

**>>Thank you for this suggestion. Indeed, references to the latest demographic census will be added and L246 will be amended as follows:**

**A notable challenge to the livelihood system, not related to drought, is the aging population of Piquet Carneiro and their purchasing power. Specifically, retirees, who predominantly purchase farmers' products in local markets, determine the sales pattern. According to census data, the ageing index (índice de envelhecimento used by the IBGE) in Piquet Carneiro increased from 66 in 2010 to 110.16 in 2022, meaning that for every 100 children aged 0 to 14, there were 110.16 adults aged 60 or older by 2022. This shift towards an ageing population is more pronounced in Piquet Carneiro compared to the state of Ceará, where the ageing index increased from 41.56 to 71.6 over the same period (IBGE, 2022).**

**Instituto Brasileiro de Geografia e Estatistica (IBGE): Censo Demográfico. Tabela 9515 - Índice de envelhecimento, idade mediana e razão de sexo da população : https://sidra.ibge.gov.br/tabela/9756 (last access: 12 Agust 2024), 2022.**

Figure 2: Provide an overall title as well as the subtitles. It isn't completely clear why all this information needs to be in the same Figure. Could it be more than one figure? Legends are difficult to read on all subgraphs. Banana yield/area colors are impossible to distinguish – please use more contrast. The tiny graphs at the bottom of SPI1s and drought colors are very small and consequently difficult to interpret. Are they part of Figure d and if so why? Also, the subgraphics are not labeled (which one is 2A, for example?).

**>>Thank you very much for this suggestion. We have experimented with various configurations for visualising the data, and grouping all the information into a single figure proved to be the most effective way to compare the evolution of the different conventional indices over the years. However, we agree that the figure could benefit from greater clarity. Therefore, we will enlarge all legends, label all sub-graphs, add a separate label for the SPI scale as an additional sub-graph (Figure 2e), and adjust the colours and contrast of the banana plots.**

Figure 3: Very difficult to interpret at the scale provided. Please enlarge. Please label the locations – impossible to make out which is the red circle and which is the black circle. Presumably also this figure presents 'surface area' rather than just area. It is the surface area of water?

**>>Thank you very much. The resolution will be improved, the red dot changed and the size of the figure extended so it is easier to distinguish the two different locations. The caption will be changed to make clear that it is the surface area of the water.**

Table 5: This is the main piece of results for the paper. It is unfortunately quite difficult to read and interpret and in some ways resembles a 'defragmentation diagram' on a computer. Suggestions to improve legibility include: 1) Use a consistent color framing for blindspots, mismatches, and matches and provide a legend 2) Distinguish different grades of blindspots and mismatches perhaps with different colors 3) Provide three separate tables rather than three sub-tables, and give each one more space 4) Provide a code that is more descriptive than 'M1' but less wordy than the descriptions at the bottom of the table. Unclear why the 'B's start with B8. Why not B1? In any event, the letter/number codes are difficult to interpret. The table overall

is important and very interesting, but it is just extremely difficult to read given that some cells have text, others have colors, there are two different colors for blindspots, and no discernible pattern to the use of codes and numbers. If the goal of this 'matrix' is for it to be used by others, it needs to be easy to read and use.

**>> Thank you for the suggestions. We really tried most of the suggestions. If we use a colour frame, there will be too many colours and the need to justify their choice with a qualitative colour scale. Instead, we will specify the legend in the subcaption for purple (blindspot), orange (misatches), and white for a match. Since we need to fit a lot of information into the table, we have limited the number of characters to 2-3 to minimise the space needed and so that it fits the page in a portrait format. 'B' represents blindspots and 'M' represents mismatches (as already detailed in the caption), with 'B' starting at 8 because 'M' ends at 7. We found it less confusing when the codes are unique, so the numbers do not repeat with the letters. It is not that some cells have codes and others have colours; all have both codes and colours. Cells left white indicate matches or N/A. To improve clarity, we will not merge the cells in this table and will add all the borders to make it easier to understand the information in each cell.**

Line 343: Really interesting results here and elsewhere that describe differences in perceptions of local people and the 'official' records.

**>>Thank you**

Line 349: Elsewhere the paper refers to '2003-2012' but here it appears as 2004-2011 – please be consistent.

**>>Thank you for pointing that out. L349 will be amended to 2003-2012**

Line 405 and surrounding paragraph: Very interesting interpretation and important highlighting of the impact of this approach.

**>>Thank you very much**

Line 468: It is a little disappointing that the paper does not provide alternative indicators. Why not? A couple could be suggested to help overcome the most important blindspots? It seems a real shame not to have included here some suggestions for improving monitoring.

**>>Thank you very much. This is true. However, we wanted to keep the focus of the paper centered around the MEM and the reasons underlying a misalignment between what is monitored and what is experienced by the community. The indices relevant to the community of Olho d'Água are revealed in the results section, and the discussion, highlighting elements like some income-generating activities and policies that increase resilience to drought. But the whole premise of the paper is that what fits one community might not fit another. We, therefore, believe that it would defeat the purpose of our study and backfire on us to suggest any indicator based on local and empirical observations that might not be backed up by a strong theoretical framework itself.**

Line 480: Humans are not machines and so yes, all interviews will be different and present a version of reality. Question whether 'biases are negligible' can be proven based on the fact that other interviewees said similar things – as nothing else has changed (questions and research set up are the same) – this is more about triangulation and corroboration rather than addressing individual people's 'bias'.

**>>Thank you. L480 will be amended as follows:**

**While these biases are inherent to the interview process and the setup of interviews can vary, we find that the overall trends identified are consistent with those observed in other communities, supporting the robustness of our findings despite these limitations.**

Line 484: Question whether a paper 'under consideration' can be used as a citation, update with actual citation if now accepted.

**>>Thank you very much. All the articles listed ' under consideration' are now published and accessible. Therefore, all will have their year if publication added.**

Line 506: Drivers are not qualitative. The way drivers are assessed might be done using qualitative methods. Some slight change in language suggested.

**>>Thank you. L506 will be amended as follows:**
**Many human factors influencing resilience and vulnerability to drought impacts are assessed qualitatively, as shown in this study.**

Line 521: Excellent and relevant conclusion.

**>>Thank you very much.**

Line 528: Is this practical? Were technical officers interviewed? Is this something that they have capacity and funding to do?

**>>Thank you very much for your suggestions. Regarding the practicality of delegating monitoring responsibilities to agricultural technicians, we would like to point out that we have addressed this aspect in Line 498, where we discuss how this approach is already being implemented at the state level in Ceará. The study we reference highlights that this type of monitoring is indeed practical and is currently being applied to some extent (Walker et al., under consideration – in the text, now corrected to Walker et al., 2024). However, regarding the financial feasibility of expanding this approach, we acknowledge that this would require a separate study to provide reliable information. Offering remarks without such an analysis would be speculative at best.**